# Emergence of supercoiling-mediated regulatory networks through the evolution of bacterial chromosome organization

**Théotime Grohens**[1], **Sam Meyer**[2], **Guillaume Beslon**[3]*

**1** LBMC, ENS de Lyon, CNRS, UMR 5239, Inserm, U1293, Université Claude Bernard Lyon 1, Lyon, France, **2** Université de Lyon, INSA Lyon, Université Claude Bernard Lyon 1, CNRS, UMR5240 MAP, France, **3** INSA Lyon, Inria, CITI/BioTiC, Lyon, France

* guillaume.beslon@insa-lyon.fr

**Data availability statement:** The source code for the simulation, as well as the notebooks used for data analysis, are available online at

## Abstract

DNA supercoiling—the level of twisting and writhing of the DNA molecule around itself—plays an important role in the regulation of gene expression in bacteria by modulating promoter activity. The level of DNA supercoiling is a dynamic property of the chromosome which varies both at local and global scales, in response to both external factors such as environmental perturbations and internal factors including gene transcription. As such, local variations in supercoiling could in theory couple the expression levels of neighboring genes by creating feedback loops at the transcriptional level. However, the impact of such supercoiling-mediated interactions on the regulation of gene expression still remains uncertain. In this work, we study how this coupling between transcription and supercoiling could shape genome organization and help regulate gene transcription. We present a model of genome evolution in which individuals whose gene transcription rates are coupled to local supercoiling must adapt to two environments that induce different global supercoiling levels. In this model, we observe the evolution of whole-genome regulatory networks that provide control over gene expression by leveraging the transcription-supercoiling coupling, and show that the structure of these networks is underpinned by the organization of genes along the chromosome at several scales. Local variations in DNA supercoiling could therefore help jointly shape both gene regulation and genome organization during evolution.

## Author summary

DNA, the carrier of genetic information, is a flexible molecule that can dynamically twist and writhe around itself, a property known as DNA supercoiling. DNA supercoiling plays a particular role in gene regulation, because it can both affect gene transcription and be affected by it in return: genes located in underwound DNA are usually

the following address: https://www.github.com/tgrohens/evotsc and on the Software Heritage archive at the following address: https://archive.softwareheritage.org/swh:1:dir:dc7169fdc35871aa456a7fecd095f7e0758bc368. The data from the main run of the experiment is available online on the Zenodo platform, at the following address: https://doi.org/10.5281/zenodo.7062757. The supplementary data is available at the following address: https://doi.org/10.5281/zenodo.17077963.

**Funding:** The author(s) received no specific funding for this work.

**Competing interests:** The authors have declared that no competing interests exist.

expressed more, and when a gene is being transcribed, DNA both overwinds downstream and underwinds upstream of the gene. In this work, we study the impact of this coupling between gene regulation and DNA supercoiling on the organization of bacterial genomes. We present a computational model in which simulated bacteria must adapt the expression of their genes, which depends only on supercoiling, to different environments by reordering their genomes through genomic inversions over generations. We show that, in this model, environment-specific gene expression can indeed evolve, and is the result of the formation of specific patterns of gene positions and orientations along the genome, leading to the emergence of supercoiling-sensitive regulatory networks. Altogether, these results suggest that gene regulation via supercoiling can help understand the organization of bacterial genomes through an evolutionary lens, and that this mechanism should be accounted for when designing fine-tuned artificial genetic constructs.

## 1. Introduction

DNA is the material basis of genetic information. It is a flexible polymer comprising two strands of nucleotides that coil around each other. In the cell, DNA is subject to torsional stress, inducing what is known as DNA supercoiling in the form of twist (deformation around the helical axis) and writhe (3D deformation of the helical axis). The level of supercoiling $\sigma$ is an important physiological parameter, and it is measured in a given DNA molecule as the relative difference between its linking number (the number of times its strands cross each other, which is affected by twist and writhe) and that of a relaxed DNA molecule of the same length. In bacteria, DNA is normally maintained in a negatively supercoiled state, with a typical value of $\sigma_{basal}$ = −0.06 in *Escherichia coli* [1]. In these organisms, DNA supercoiling is an important regulator of gene expression, as changes in the level of supercoiling directly affect gene transcription rates [2–4].

**Regulation of DNA supercoiling.** In bacteria, the level of DNA supercoiling is primarily controlled by topoisomerases, enzymes that alter supercoiling by passing DNA through single- or double-strand breaks. The two main topoisomerases are gyrase, which introduces negative supercoils at an ATP-dependent rate, and topoisomerase I, which oppositely relaxes negative supercoiling without ATP [5]. The average chromosomal supercoiling level depends on their relative activity, and changes depending on environmental conditions. A prime example can be found in *E. coli*, where DNA is more negatively supercoiled during exponential growth, and more relaxed in stationary phase [1]. Supercoiling also responds to environmental stresses: salt shock causes a transient increase in negative DNA supercoiling in *E. coli* [6]; an acidic intracellular environment causes DNA relaxation in the facultative pathogen *Salmonella enterica* var. Typhimurium [7]. In the plant pathogen *Dickeya dadantii*, higher temperatures also relax DNA, and lead to the activation of a virulence gene [8], suggesting that supercoiling could mediate the expression of pathogenic genes in response to stress in this bacteria.

These responses are usually very well conserved among species [9], and the level of supercoiling therefore constitutes a global regulatory parameter subject to cellular control. Moreover, supercoiling does not only vary through time, but also depending on chromosomal location. In particular, according to the *twin-domain* model of supercoiling presented in [10], the transcription of a gene by an RNA polymerase generates both positive supercoils downstream and negative supercoils upstream of the transcribing polymerase [11–13], as a consequence of the drag that hampers the rotation of the transcriptional complex (polymerase,

RNA, and associated ribosomes and proteins) around DNA during transcription. Furthermore, while the intrinsic flexibility of the DNA polymer allows supercoils to diffuse along the chromosome, physical constraints induced by the 3D confinement of the nucleoid and by DNA-bound proteins form topological obstacles and barriers that block its diffusion [14–16]. As a result, bacterial chromosomes are estimated to be partitioned in topological domains with an average size of 10 kb [14].

**Global regulatory role of DNA supercoiling.** The level of DNA supercoiling influences gene expression in bacteria through several mechanisms [2–4]. In particular, more negatively supercoiled DNA facilitates the initiation of gene transcription, as opening the DNA double strand–the initial step of gene transcription–is thermodynamically favored in more negatively supercoiled DNA regions [17]. In *E. coli*, [3] showed that 7% of genes were either up- or down-regulated by a global relaxation (a less negative supercoiling level) of chromosomal DNA. Similar transcriptional responses to DNA relaxation were obtained for *S. enterica* [18], and for *S. pneumoniae* [19], and to an increase in negative supercoiling in *D. dadantii* [20].

DNA supercoiling might play an especially important regulatory role in bacteria with reduced genomes, such as the obligate aphid endosymbiotic bacterium *Buchnera aphidicola*. As *B. aphidicola* is nearly devoid of transcription factors, global and local changes in supercoiling are thought to be one of the main mechanisms available to this bacteria for the regulation of gene expression [21]. In *Mycoplasma pneumoniae*, another genome-reduced bacterium, other work has also shown that supercoiling plays a global regulating role, and that it might mediate transcriptional interference between convergent genes [22]. Finally, mutations that alter the regulation of DNA supercoiling have also been shown to be evolutionarily favorable in experimental settings. For example, in the so-called *Long-Term Evolution Experiment*, 12 populations of *E. coli* cells seeded from a common ancestor have been adapting to a laboratory environment for over 75,000 generations [23,24]. In this experiment, 10 out of the 12 populations were shown to present a higher level of negative supercoiling relative to their ancestor, with one population in particular presenting increase of more than 17% in negative supercoiling [25]. In this population, two distinct mutations were shown to increase negative supercoiling when inserted back into the ancestral strain, and to provide a significant growth advantage (that is, higher fitness) relative to the ancestral strain. This evolutionary trajectory was then shown to be repeatable not only at the phenotypic level, but also at the genetic level, with a majority of strains harboring mutations in DNA supercoiling-related genes [26]. In the context of the *LTEE*, supercoiling mutations therefore played a role in the adaptation of *E. coli* strains to new environments, evidencing the important effect of DNA supercoiling on bacterial gene regulation.

**The transcription-supercoiling coupling.** When two genes are located closely enough on the genome, the supercoiling generated by the transcription of one gene can in theory affect the transcription level of the other gene, and vice versa. This interaction has long been suspected to play a regulatory role [27,28], and, following [29], we will refer to it as to the *transcription-supercoiling coupling*. This coupling can take several forms depending on the relative orientation of the genes: divergent genes could increase their respective transcription level in a positive feedback loop; convergent genes could inhibit the transcription of one another; and, in tandem genes, the transcription of the downstream gene could increase the transcription of the upstream gene, and the transcription of the upstream gene decrease the transcription of the downstream gene.

Such supercoiling-mediated interactions between neighboring genes have been experimentally documented in several bacterial genetic systems. In the *E. coli*-related pathogen *Shigella flexneri*, the *virB* promoter is normally only active at high temperatures, but can be activated at low temperatures by the insertion of a phage promoter in divergent orientation [30].

Similarly, the expression of the *leu-500* promoter in *S. enterica* can be increased or decreased by the insertion of upstream transcriptionally active promoters, depending on their orientation relative to *leu-500* [31]. The transcription-supercoiling coupling has also been explored in a synthetic construct in which the inducible *ilvY* and *ilvC E. coli* promoters were inserted on a plasmid in divergent orientations [32]. In that system, a decrease in the activity of *ilvY* was associated with a decrease in *ilvC* activity, and an increase in *ilvY* activity with a corresponding increase in *ilvC* activity. More recent work using plasmids in *E. coli* has shown that induction of a divergently oriented gene can result in lower expression of a target gene through slower transcription [33], or that a convergent or tandem orientation can result in higher expression of a target gene than a divergent orientation [34,35], suggesting that the twin-domain model can however not always be straightforwardly used to make quantitative predictions.

The biological relevance of the transcription-supercoiling coupling might however not be limited to such local interactions. Indeed, in *E. coli*, the typical size of topological domains– inside which the positive and negative supercoils generated by gene transcription can propagate–is usually estimated to range around 10 kb [14], and transcription-generated supercoiling has been shown to propagate up to 25 kb in each direction around some specific genes [13]. As genes stand on average 1 kb apart on the *E. coli* chromosome [36], any single topological domain could therefore encompass multiple genes interacting via the transcription-supercoiling coupling. Supercoiling-sensitive genes have indeed been shown to group in local up- or down-regulated clusters, found all around the chromosome, in bacteria such as *E. coli* [3], *S. enterica* [18] and *S. pneumoniae* [19]. In *E. coli*, a statistical analysis of the relative position of neighboring genes on the chromosome indeed showed that genes that are up-regulated by a global increase in negative supercoiling have more neighbors in divergent orientations, while genes that are down-regulated in these conditions have more neighbors in convergent orientations [37]. The co-localization of genes in such clusters has therefore been hypothesized to play a phenotypic role by enabling a common regulation of their transcription through local variations in the supercoiling level. At a larger scale, recent advances in long-read sequencing have facilitated the detection of structural rearrangements in bacterial genomes and the study of their impact on gene expression. In a long-term growth experiment using *S. enterica*, genomic inversions were found to affect the expression level of genes in the inverted region, with genes at the extremities of the inversion being affected the most [38]. In particular, one of these inversions evolved twice during the experiment, suggesting it provided an evolutionary advantage in that context. While proximity to the origin of replication and co-orientation with the replication machinery are important drivers of differential expression of genes along the chromosome, supercoiling-mediated gene expression variation at inversion boundaries could also be partially responsible for the observed fitness gains due to the inversion. In *E. coli*, prophage-mediated genomic inversions were also detected in both lab and cattle strains, leading to increased virulence (and possible evolutionary advantage) in some of these strains [39]. Finally, synteny segments – clusters of neighboring orthologous genes that show correlated expression patterns – have been shown to be more conserved than expected by chance in a study considering over 1,000 bacterial species, including in particular the distantly related *E. coli* and *Bacillus subtilis*, possibly as a consequence of co-regulation of the genes within these segments through supercoiling [40]. Overall, this body of empirical evidence suggests that local variations in the supercoiling level, due to its coupling with transcription, could indeed play a substantial role in the regulation of gene activity and consequently impact the evolution of genome organization.

Multiple modeling works have recently addressed the simulation of the transcription-supercoiling coupling, reporting contrasted results [17,41–44]. Quantitative regulatory rules

that result from this coupling thus still remain to be fully characterized. Here, we address a different question that has never been subject to detailed analysis: how the transcription-supercoiling coupling may drive genome evolution. We present a two-level framework, in which a whole-genome model of the transcription-supercoiling coupling is embedded within an evolutionary simulation. The regulatory aspect of the model is voluntarily kept simple, assuming that local negative supercoils always activate gene expression. In this framework, individuals must evolve gene expression levels that are adapted to two environments, characterized by different global supercoiling levels, using chromosomal rearrangements only. We first show that complex environment-driven patterns of gene expression are able to evolve in such a model, observing in particular the emergence of relaxation-activated genes. We then characterize the spatial organization of genes along the genome that is responsible for these expression patterns in the model, showing that genes are locally organized in patterns which leverage the transcription-supercoiling coupling for either activation or inhibition, such as toggle switches, but that larger-scale networks are required to strongly inhibit genes. Finally, we show that, in our model, genes form a densely connected genome-wide interaction network, overall demonstrating that supercoiling-based regulation could indeed coevolve with genome organization in bacterial genomes.

## 2. Results

### 2.1. Evolution of gene regulation through the transcription-supercoiling coupling

We introduce a model (detailed in the Methods section) in which populations of individuals must evolve to adapt their gene expression levels to two different environments encountered during their lifetime. In the model, individuals are described by a circular genome containing a constant number of genes, placed at different positions and orientations along the genome. The main assumptions of the model are that the expression level of a gene only depends on the local level of supercoiling at the location of its promoter, and that this local supercoiling level itself only depends on the following elements: the basal level of supercoiling of the chromosome, a constant perturbation of this level in each environment, and the transcription rate of neighboring genes, with a magnitude linearly decreasing with the distance to the transcribed gene (see Equations 1 to 4 in Methods). Moreover, we assume that gene transcription and DNA supercoiling are continuous functions of one another, and we assume that they always reach a deterministic equilibrium state. We can thus define, for a given individual in a given environment, a unique expression level for each gene and a unique level of supercoiling at every genome position (see Fig 12 for an example individual illustrating the model).

The two environments used in the simulation are characterized by their respective impact $\delta\sigma_{env}$ on the background supercoiling level of the chromosome of each individual. The first environment, named environment A, induces a global relaxation of DNA as in the acidic macrophage vacuoles encountered by *S. enterica* [7]. In the model, we consider that this relaxation decreases baseline gene expression ($\delta\sigma_{env} = \delta\sigma_A = 0.01$). The second environment, named environment B, oppositely induces an increase in negative DNA supercoiling, as observed for example when shifting *E. coli* cultures to a salt-rich medium [6]. In the model, we consider this negative supercoiling to oppositely increase baseline gene expression ($\delta\sigma_{env} = \delta\sigma_B = -0.01$). These environments could also be considered to reflect the stationary phase of *E. coli*, in which DNA is more relaxed, versus the exponential growth phase in which it is more negatively supercoiled [1]. In order to have high fitness, individuals must display environment-specific gene expression patterns, obtained by the activation or inhibition of three disjoint subsets of their genes–called *A*, *B* and *AB*–in each environment: *A*

genes must be activated in environment A but not in environment B, *B* genes in environment B but not in environment A, and *AB* genes must be activated in both environments. At every generation of the simulation, we compute the fitness of every individual based on their gene expression levels in each of these environments, then create the next generation by making individuals reproduce proportionally to their fitness, and applying random mutations to their offspring. Importantly, the only mutational operator that we use is genomic inversions, as our focus is the evolution of genome organization. In the model, inversions do not affect genes directly, but only change their relative positions and orientations along the chromosome. Due to supercoiling-mediated interactions, this affects gene expression levels, and in turn fitness.

**Evolution of environment-specific gene expression levels.** Using the model presented above, we let 30 populations of 100 individuals, each possessing 20 genes of each type for a total of 60 genes, evolve for 1,000,000 generations. The genome of a typical individual at the end of the evolutionary simulation is depicted in Fig 1, in environments A (left) and B (right). In each case, the outer ring depicts gene position, orientation, type, and activation, and the inner ring the local level of supercoiling due to gene transcription $\sigma_{TSC}$. For ease of interpretation, we discretize expression levels by considering that a gene is activated if its expression is above the expression threshold $e_{1/2}$, and inhibited otherwise (see Methods). In this individual's genome, different activation patterns are visible, for each gene type and as a function of the environment. All the *AB* genes except one (gene 51) are correctly activated (dark blue) in the two environments; 19 out of 20 *B* genes are correctly inhibited (light green) in environment A (left) while 18 are correctly activated (dark green) in environment B (right); and 16 *A* genes are activated (dark red) in environment A, while 16 are inhibited (light red) in environment B. Note that this asymmetry between the number of *A* and *B* genes that are in the

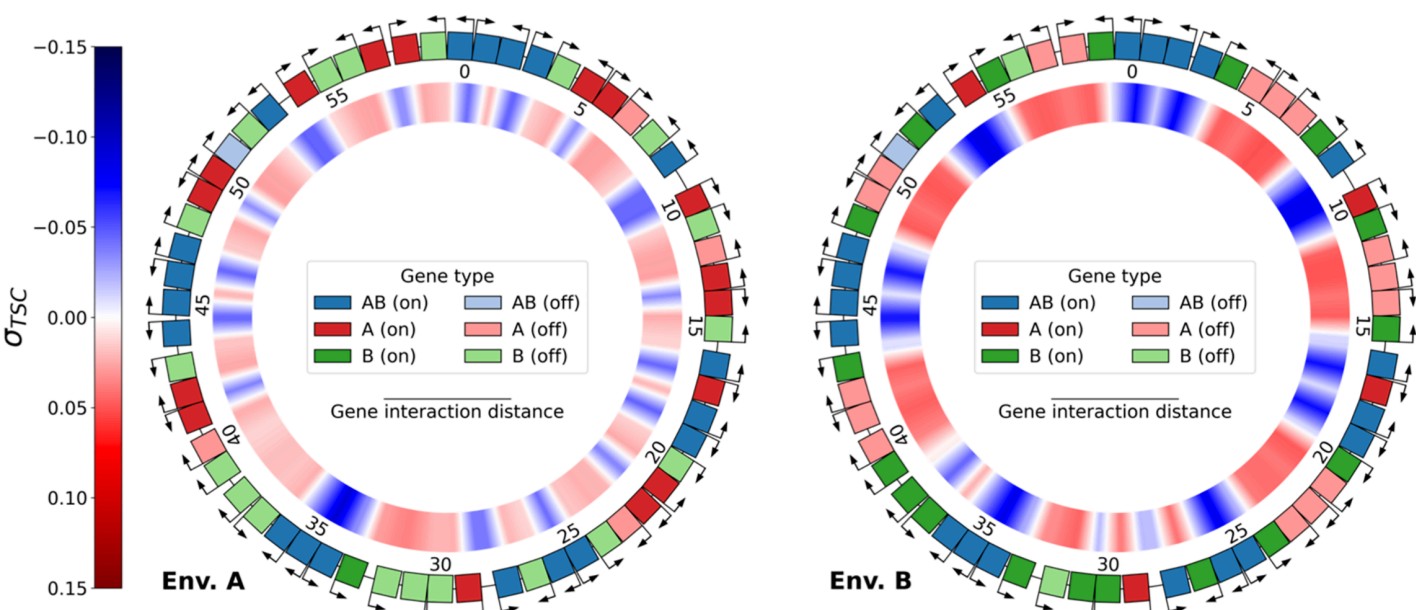

**Fig 1. Genome of the best individual at the last generation of evolution of replicate 21, evaluated in the two environments: A (relaxed DNA, left) and B (more negatively supercoiled DNA, right).** The outer ring shows the type, orientation, and expression of each gene on the genome (darker color: activated; lighter color: inhibited). Genes are considered to be activated if their expression is above the threshold $e_{1/2}$, and inhibited otherwise (see Methods). Genes are numbered clockwise according to their position on the genome. The inner ring shows the level of transcription-generated DNA supercoiling $\sigma_{TSC}$ at every position on the genome. Shades of blue represent negative supercoiling ($\sigma_{TSC} < 0$), and shades of red positive supercoiling ($\sigma_{TSC} > 0$).

expected state is due to an asymmetry in the effect of the environments themselves, which we will discuss below in more detail.

The transcription-generated supercoiling that is represented in the inner ring can also be seen to change consistently with the gene activation patterns between the two environments: zones of negative DNA supercoiling (in blue) are delineated by divergently oriented activated genes, while zones of positive DNA supercoiling (in red) contain inhibited genes. The genome of this evolved individual therefore shows that, in the context of this model, it is possible for evolution to adjust gene expression levels to an environment-dependent target solely by rearranging relative gene positions and leveraging the transcription-supercoiling coupling between neighboring genes.

This behavior is however not specific to this particular individual. Fig 2 shows that the fitness of the best individual of each population, averaged over all populations, evolves smoothly towards higher values over the course of the simulation. More precisely, Fig 3 shows that the

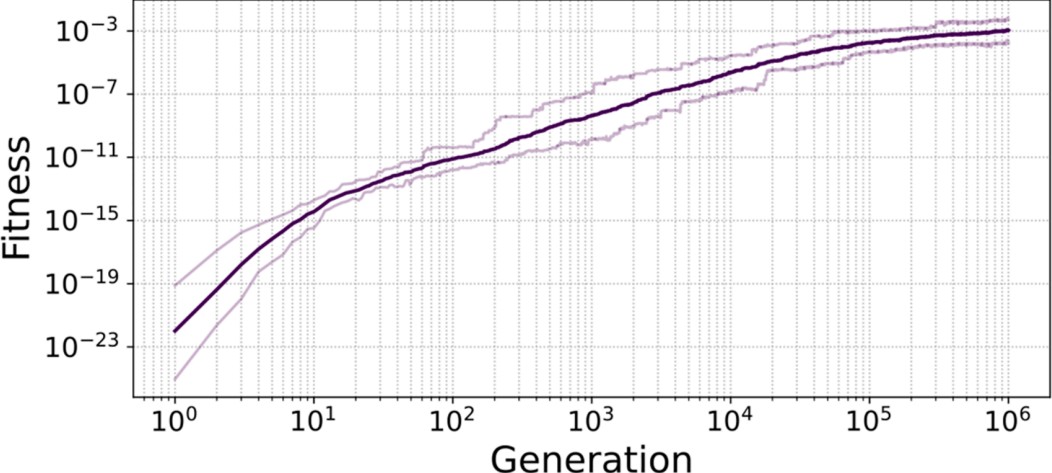

**Fig 2. Geometric average of the fitness of the best individual in each of the 30 populations, at every generation.** Lighter lines represent the first and last decile of the data.

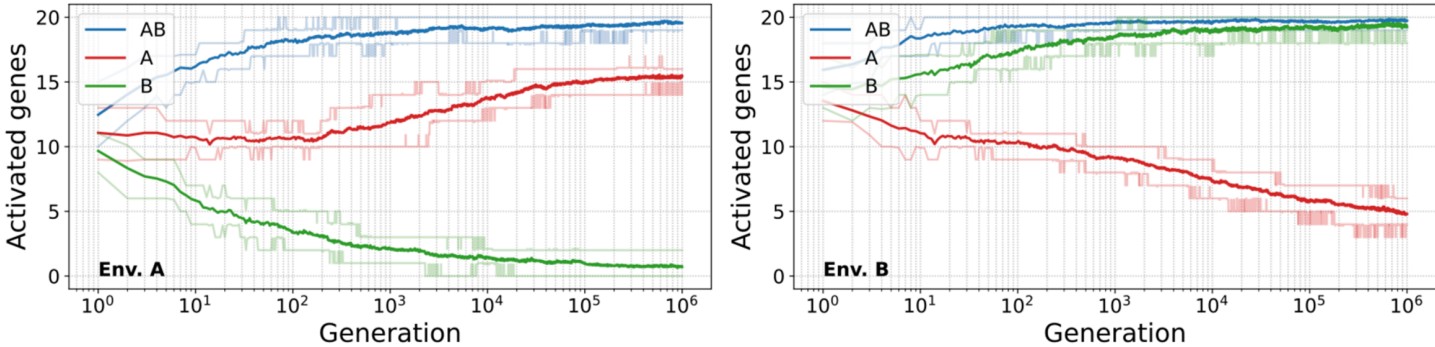

**Fig 3. Average number of activated genes (expression higher than $e_{1/2}$) of each type in the best individual at every generation, averaged over the 30 populations, in environments A (left) and B (right).** Lighter lines represent the first and last decile of the data.

numbers of activated genes of each particular type also evolve towards their respective targets, in environment A (left) and B (right). In each environment, the average number of activated *AB* genes (in blue) quickly reaches nearly 20, its maximum value, as expected from their target; *B* genes (in green) show the same behavior, evolving towards nearly full activation in environment B and nearly full inhibition in environment A. *A* genes (in red) follow a slightly different course, as the number of activated *A* genes seems to converge to approximatively 15 out of the expected 20 in environment A, but continues to decrease towards the expected 0 in environment B by the end of the simulations.

The incomplete match of *A* genes with their evolutionary target–as compared to *B* genes–could however be partially expected. Environment A is indeed characterized by a *less negative* global supercoiling level, while environment B is characterized by a *more negative* global supercoiling level (recall that the basal supercoiling level is negative). As less negative supercoiling reduces gene transcription, it is by construction more difficult for a gene to have a high transcription rate in environment A than in environment B. *A* genes must therefore complete the more difficult task of being activated in the inhibiting environment A, while being inhibited in the activating environment B, whereas *B* genes must complete the comparatively easier task of being activated in the "easier" environment B and inhibited in the "harder" environment A.

Well-differentiated expression levels nonetheless do evolve in our model for both types of genes, in response to the different supercoiling levels imposed by the environmental conditions. These gene expression patterns are moreover remarkably robust to the magnitude of the difference between the environments. Indeed, we still observe such patterns even when repeating the experiment with environmental perturbations 10 and 100 times smaller in magnitude (see Figs A and B in S1 Text).

**Evolution of relaxation-activated genes.** In our model, the expression of a gene that doesn't interact with its neighbors increases with the amount of negative supercoiling at its promoter (see Equations 3 and 4 in Methods). One could therefore expect that the genes of evolved individuals would present a qualitatively similar response. In order to verify this, we measured the expression of every gene in the genomes of evolved individuals, not only in the two discrete environments characterized by $\delta\sigma_A$ and $\delta\sigma_B$ (as throughout the evolutionary process), but instead as a function of a continuously varying $\delta\sigma_{env}$. Fig 4 presents this data, grouping genes by their type, and averaging over the 30 replicates.

Fig 4 highlights striking differences between the average response of each type of gene in evolved genomes (red, green, and blue lines) on the one hand, and the responses of isolated, non-interacting genes (dashed light blue line) or of genes in random non-evolved genomes (dash-dotted light blue line) on the other hand. While *AB* and *B* genes (in blue and green respectively) display an average expression level that decreases as supercoiling increases, and that remains qualitatively similar to the behavior of random genes (dash-dotted line), *A* genes (in red) display a completely different behavior. Indeed, *A* genes show a non-monotonic response to the environmental perturbation in supercoiling, as their average expression level decreases until a local minimum in expression at $\delta\sigma_B$, then increases–even though background negative supercoiling decreases–until a local maximum at $\delta\sigma_A$, before decreasing again similarly to other genes types. In other words, due to their interaction with other genes, *A* genes present a phenotype of activation by environmental relaxation of DNA for perturbations between $\delta\sigma_B$ and $\delta\sigma_A$, even though the activity of an isolated *A* gene would decrease with background DNA relaxation (dashed light blue line). The evolution of this relaxation-activated phenotype is furthermore very robust to the magnitude of the environmental perturbations, as we can again observe it when replaying the main experiment with supercoiling

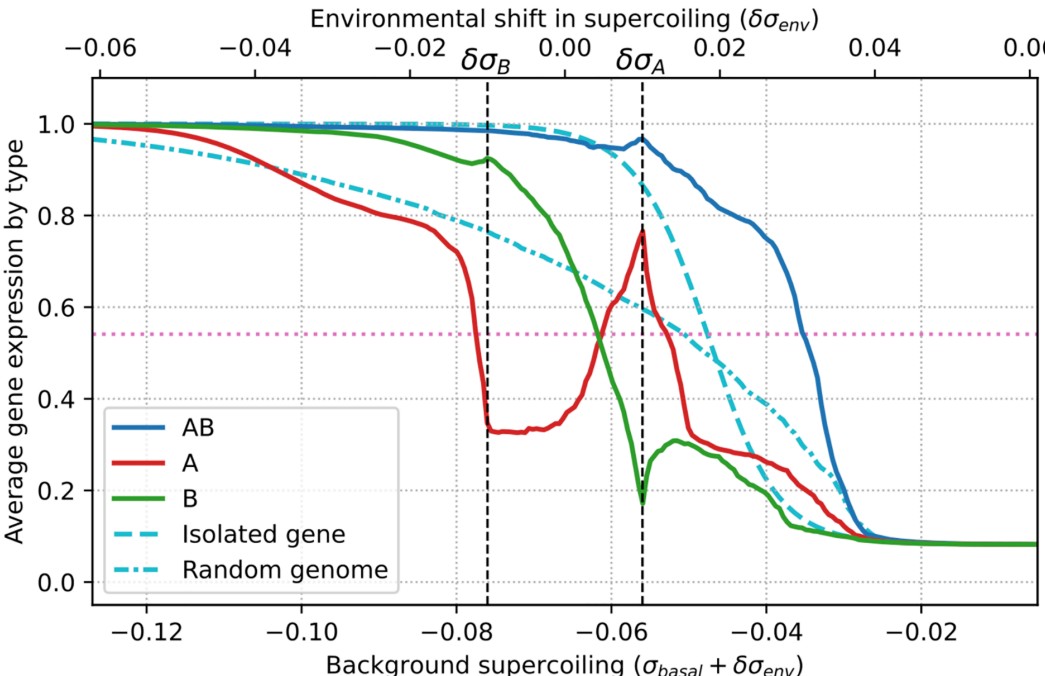

**Fig 4. Average gene expression level per gene type as a function of the environmental perturbation in supercoiling.** The full colored lines show average expression levels for each gene type (*AB*, *A*, and *B* in blue, red, and green respectively), as a function of the perturbation $\delta\sigma_{env}$ (top horizontal axis), or equivalently of the background supercoiling level $\sigma_{basal} + \delta\sigma_{env}$ (bottom horizontal axis), averaged over the best individual in each of the 30 populations. The dashed light blue line represents the expression level of a single neighbor-less gene, and the dash-dotted light blue line represents the average expression level of genes on 30 random genomes. The dashed vertical lines represent the perturbations $\delta\sigma_A$ and $\delta\sigma_B$ due to environments A and B, in which individuals evolved during the simulation, and the pink horizontal line marks $e_{1/2}$, the threshold above which a gene is considered active.

perturbations that are 10 or 100 times smaller than in the main experiment (see Figs C and D in S1 Text respectively).

In our model, the transcription-supercoiling coupling is therefore able to provide a regulatory layer that mediates the transcriptional response to the global variation in DNA supercoiling due to environmental perturbations. Remarkably, the coupling allows for the evolution of activation by DNA relaxation, meaning that a global decrease in negative supercoiling can lead to a localized increase in negative supercoiling, and therefore demonstrating the importance of relative gene positions on the regulation of transcription.

## 2.2. Evolution of local genome organization

Having first shown that differentiated gene transcription patterns can evolve in our simulations in response to environmental perturbations, we then sought to determine the genome organization that necessarily underlies these patterns in our model, given that the only difference between individuals in the model is the relative position and orientation of the genes on their genomes.

In order to do so, we systematically enumerated all possible pairs and triplets of neighboring genes of different types and in different relative orientations, and measured their abundances in evolved genomes, which we will describe starting with gene pairs. As the interaction in a gene pair can be asymmetric if the genes are in tandem, we gave gene pairs an orientation by considering that a pair consists of a focal gene which affects a target gene. We first measured the abundance of such oriented gene pairs in each relative orientation (convergent, divergent, focal gene upstream, or focal gene downstream), as the relative orientation of two neighboring genes determines their mode of interaction through the transcription-supercoiling coupling: mutual activation for divergent genes, mutual inhibition for convergent genes, and activation (resp. inhibition) of the upstream (resp. downstream) gene by the downstream (resp. upstream) gene. As, in our model, different gene types must evolve different activation patterns in each environment for an individual to have high fitness, we further stratified these pair counts by the type of each gene in the pair, resulting in 9 kinds of oriented pairs. Finally, in order to quantify the actual strength of the coupling between the genes in a given type of pair, we also measured the total level of positive and negative supercoiling generated by the transcription of the focal gene at the promoter of the target gene, for all relative orientations. These data are presented in Fig 5.

**Frequent gene pairs in evolved genomes showcase twin-domain behaviors.** The most frequent pair pattern found in evolved genomes is that of a convergently oriented $\vec{A} - \overleftarrow{B}$ pair

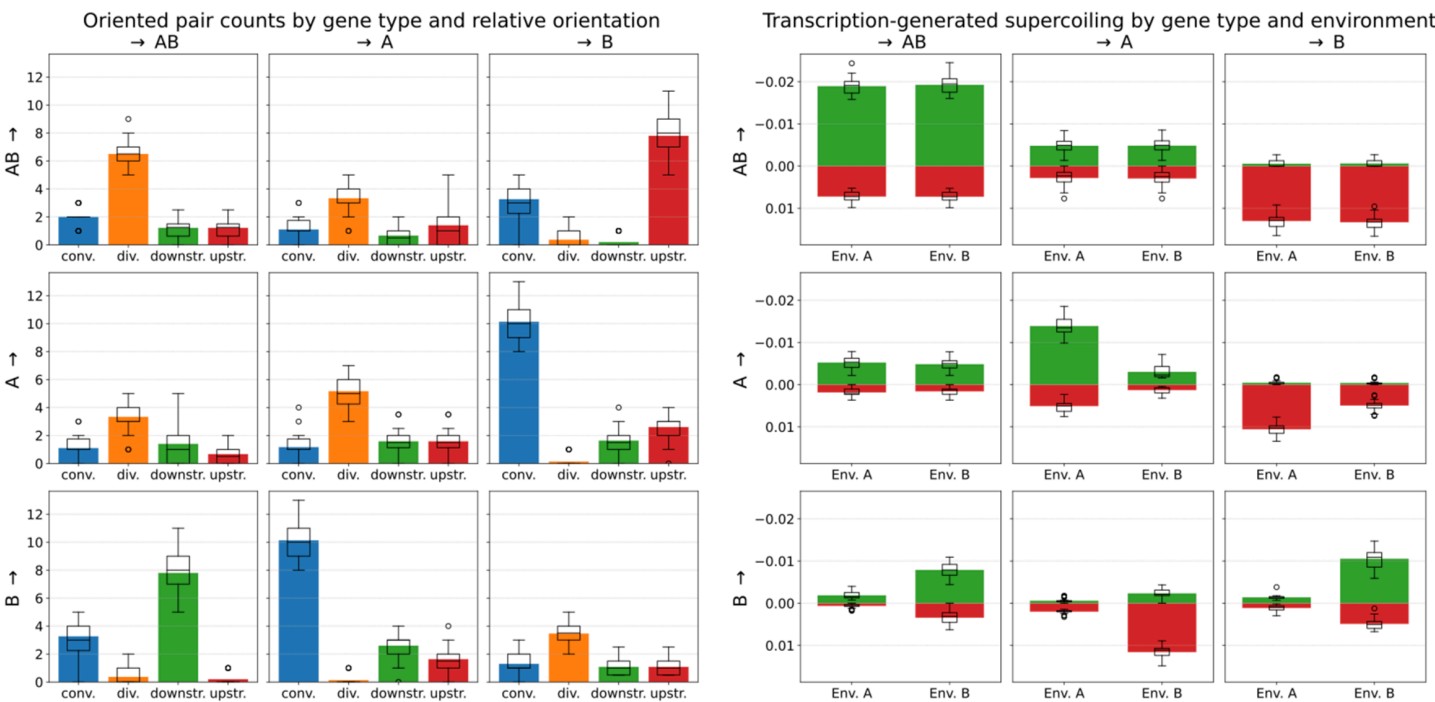

**Fig 5. Interactions in oriented pairs of neighboring genes.** The left-hand side panel shows the number of oriented pairs of each kind, split by the type of the focal gene (row) and of the target gene (column) in the pair, and by relative orientation (bars in each sub-panel: convergent, divergent, focal upstream, or focal downstream). For instance, the $AB \rightarrow B$ top-right panel shows the influence of $AB$ genes on $B$ genes, and the $B \rightarrow AB$ bottom-left panel the influence of $B$ genes on $AB$ genes (in the same pairs). In these pairs, there are on average 7.8 $AB$ genes directly upstream of a $B$ gene (top-right panel, in red), or equivalently 7.8 $B$ genes directly downstream of an $AB$ gene (bottom-left panel, in green) on an evolved genome. The right-hand side panel shows, for each kind of oriented pair, the total amount of negative (green) and positive (red) transcription-generated supercoiling due to the focal type (row) measured at the promoter of the target type (column), summed over all orientations, and split by environment. All data is averaged over the final best individual of each of the 30 replicates. Box plots indicate the median and dispersion between replicates, and circles denote outliers.

(or, equivalently, a $\vec{B} - \overleftarrow{A}$ pair, using overhead arrows to denote relative gene orientation). There are on average just over 10 such pairs in an evolved genome (blue bars in the $A \to B$ and $B \to A$ panels on the left-hand side of Fig 5). In such pairs, according to the twin-domain model of supercoiling, each gene theoretically inhibits the expression of the other gene in the pair through the downstream generation of positive supercoiling. The right-hand side panel of Fig 5 shows that, in environment A, $A$ genes indeed generate an average positive supercoiling variation of 0.01 at the promoter of convergently oriented $B$ genes, and hence decrease the expression of such $B$ genes with a strength that is comparable to the supercoiling perturbation due to environment A ($\delta\sigma_A = 0.01$). In this environment, $B$ genes are mostly inhibited, and therefore do not strongly impact the expression of $A$ genes. In environment B, it is oppositely $B$ genes that strongly inhibit the expression of convergently oriented $A$ genes, through the generation of positive supercoiling at the promoter of such $A$ genes. In evolved genomes, convergent $\vec{A} - \overleftarrow{B}$ gene pairs therefore seem to behave as toggle switches, or bistable gene regulatory circuits, in which the expression of one gene represses the expression of the other gene [45].

The second most frequent pattern is that of $AB$ genes placed immediately upstream of a $B$ gene, or $\overrightarrow{AB} - \vec{B}$ pairs, with on average just under 8 such pairs found in an evolved genome. This can be again understood in terms of the expression target of the genes in the pair: as visible in the right-hand side panel of Fig 5, $AB$ genes in such pairs generate a large amount of positive supercoiling at the downstream $B$ gene, which is needed for inhibition in environment A, and compensated by the negative supercoiling perturbation of environment B.

Finally, divergently oriented pairs of all three kinds of genes are also frequently found in evolved genomes. In particular, divergent $\overleftarrow{AB} - \overrightarrow{AB}$ gene pairs generate an average negative supercoiling of around −0.012 at their promoters, in both environments (summing the positive and negative bars in the $AB \to AB$ sub-panel on the right-hand side of Fig 5). This value is comparable in magnitude to, but has the opposite sign than, the supercoiling perturbation due to environment A ($\delta\sigma_A = 0.01$). The interaction between neighboring genes can therefore locally counteract the global shift in supercoiling caused by this environment, in order to maintain environment-agnostic high gene expression levels. This unconditionally positive feedback loop would, on the contrary, seem less evolutionarily favorable for $\overleftarrow{A} - \vec{A}$ or $\overleftarrow{B} - \vec{B}$ pairs than for $\overleftarrow{AB} - \overrightarrow{AB}$ pairs, as both $A$ genes and $B$ genes must be conditionally expressed or inhibited depending on the environment. We can indeed observe that divergent $\overleftarrow{A} - \vec{A}$ and $\overleftarrow{B} - \vec{B}$ pairs result in slightly weaker interactions (middle and bottom-right sub-panel of the right-hand side of Fig 5), in the environment in which these genes are active. In contrast to these frequent pairs, one can note that divergent $\overleftarrow{A} - \vec{B}$ gene pairs are almost never found, consistently with theoretical expectation, since $A$ and $B$ genes must not be expressed in the same environment.

**Gene triplet frequencies reflect pairwise interactions.** Following the same approach as for pairs, we then measured the observed frequencies of gene triplets, stratifying triplets by the type and orientation of each gene they contain, which results in $3^3 * 2^3 = 216$ possible triplets. We grouped symmetric triplets together (for example, $\overrightarrow{AB} - \vec{B} - \overleftarrow{A}$ together with $\vec{A} - \overleftarrow{B} - \overleftarrow{AB}$), keeping by convention triplets whose central gene is in leading orientation, thus reducing the number of triplets under consideration to 108.

Fig 6 shows the gene triplet frequencies observed in the genomes of the best individuals at the end of the simulations, sorted by gene type and orientation (top heatmap) or by decreasing frequency (bottom bar and box plots). As in the case of pairs, triplet frequencies strongly depend on their gene content: in particular, three specific triplets stand out as much more frequent than all others. These triplets are visible as yellow-green squares in the heatmap at the

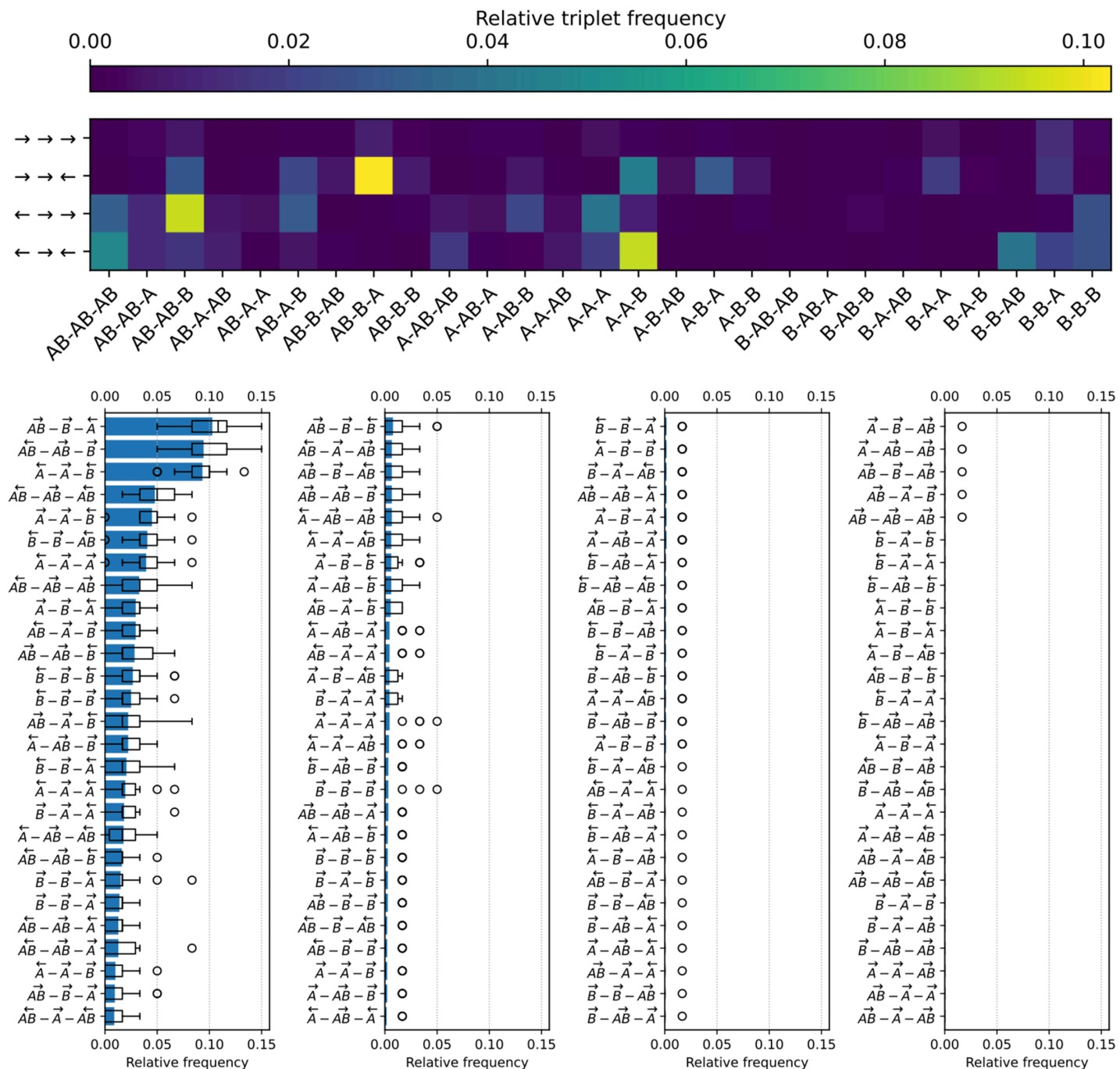

**Fig 6. Relative frequencies of gene triplets.** Top: heatmap of relative gene triplet frequencies, with gene types on the X-axis and relative gene orientations on the Y-axis. Note that we count symmetric triplets together, keeping by convention the triplet whose central gene is in leading orientation. Bottom: gene triplets sorted by decreasing relative frequency (top to bottom and left to right), with relative gene orientations depicted as overhead arrows. All data is averaged over the best final individuals of the 30 replicates. Box plots indicate the median and dispersion between replicates, and circles denote outliers.

top of Fig 6: by decreasing frequency, $\overrightarrow{AB} - \overrightarrow{B} - \overleftarrow{A}$ (1), $\overleftarrow{AB} - \overrightarrow{AB} - \overrightarrow{B}$ (2), and $\overleftarrow{A} - \overrightarrow{A} - \overleftarrow{B}$ (3). The fact that these are the most frequent triplets is consistent with the previously discussed distribution of frequent gene pairs, as each of these triplets contains either a divergent $\overleftarrow{AB} - \overrightarrow{AB}$

pair or a convergent $\overrightarrow{A} - \overleftarrow{B}$ pair, and allow us to gain further insight about the regulation of these pairs. Indeed, triplet (1) shows how the $B$ gene of a convergent $\overrightarrow{A} - \overleftarrow{B}$ pair can be under inhibitory control of an $AB$ gene (remember that, with a bacteria-like negative level of basal supercoiling, it is harder to inhibit than to activate gene expression). Triplet (2) shows how an $AB$ gene in a strongly-expressed divergent $\overleftarrow{AB} - \overrightarrow{AB}$ pair can inhibit a $B$ gene, and the $B$ gene of the triplet can also be seen as reinforcing the expression of the divergent $\overleftarrow{AB} - \overrightarrow{AB}$ pair in environment B. Finally, triplet (3) shows how the $A$ gene of a convergent $\overrightarrow{A} - \overleftarrow{B}$ pair can be activated by a divergent $A$ gene, and shows a possible synergy between overlapping divergent $\overleftarrow{A} - \overrightarrow{A}$ and convergent $\overrightarrow{A} - \overleftarrow{B}$ gene pairs. Similarly to gene pairs, it is noteworthy that 22 out of 108 possible triplets never appear at all in evolved genomes (rightmost subplot of Fig 6), and an additional 17 triplets appear only once, suggesting that those triplets are under strong negative selection.

In the model, specific local genome organization patterns can therefore be evolutionarily selected or counter-selected in order to attain favorable gene expression levels through supercoiling-mediated interactions. In particular, we observed the formation of divergent gene pairs or convergent toggle switches, that either regulate or are under the regulation of their neighboring genes. However, there is no reason why phenotypically important interactions should be limited to pairs or triplets of genes, as there are always more neighboring genes just outside a given pair or triplet. As a systematic exploration would become combinatorially intractable and statistically meaningless for wider gene neighborhoods, we then studied how the behavior of a given gene depends on its neighbors at all distances.

### 2.3. Gene inhibition requires medium-range interactions

As we just saw, the local organization of the genome seems to play an important role in the regulatory response to environmental changes in supercoiling in the model. However, this local response cannot suffice to completely explain the environment- and gene-type- specific activation patterns that we observed above. Indeed, in the dense bacteria-like genomes of individuals in our model, genes do not only interact through supercoiling with their closest neighbors, but also with genes located further away on the genome. In order to quantify more precisely the range at which regulatory interactions take place in evolved genomes, we therefore studied how gene expression changes when isolating genes into contiguous neighborhoods of increasing sizes.

We applied the following algorithm: for every gene on a given genome, and for every odd subnetwork size $k$ between 1 and the genome size, we extracted the subnetwork of $k$ consecutive genes centered around that gene. Then, for each such subnetwork, we computed the expression level of every gene in the subnetwork (in the same way as for a complete genome) in each environment, and compared the activation state of the central gene with its state in the complete genome. This allowed us to compute the following metric: the minimum subnetwork size at which a gene presents the same activation state (that is, activated or inhibited) as in the complete genome, in each environment. We interpret this metric as an indicator of the complexity of the interaction network required to produce the behavior of that particular gene in that environment. Two representative examples of this algorithm are presented in Fig 7, and complete data are then shown in Fig 8.

Fig 7 depicts the smallest subnetworks that are required in order to obtain the inhibition of a representative gene of type $B$ in environment A (top row, gene 31), and of a representative gene of type $A$ in environment B (bottom row, gene 6), both taken from the genome of the same evolved individual as in Fig 1. The central $B$ gene in this example is not inhibited by a subnetwork of size 3, but requires a subnetwork of size 5 in order to be inhibited, and,

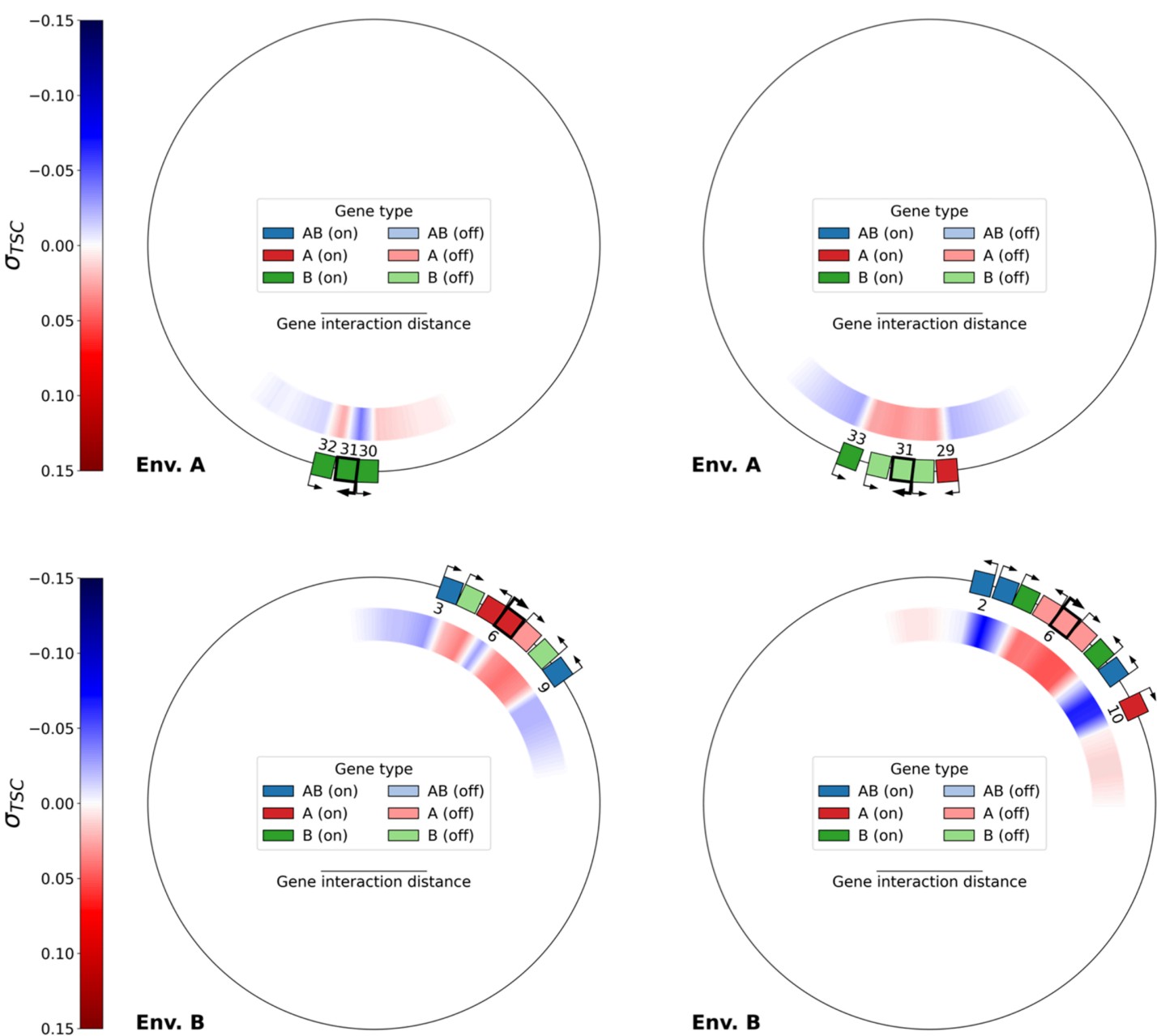

**Fig 7. Top: contiguous subnetworks of size 3 (left) and 5 (right) centered around gene 31 (of type *B*, in bold) of the final best individual of replicate 21, evaluated in environment A.** Bottom: contiguous subnetworks of size 7 (left) and 9 (right), centered around gene 6 (of type *A*, in bold) of the same individual, evaluated in environment B. In each case, when moving from the smaller to the larger subnetwork, the activation state of the central gene switches from activation to inhibition, which is the original state of these genes in the complete genome of this individual (shown in Fig 1).

similarly, the central *A* gene is not inhibited by a subnetwork of size 7, but requires a subnetwork of size 9 in order to be inhibited. In each case, increasing the size of the subnetwork by two (one gene on each side) drastically changes the expression level of the central gene, alongside with the associated level of transcription-generated supercoiling. Indeed, in the two subnetworks centered around the *B* gene (top), all 3 genes in the small subnetwork switch from

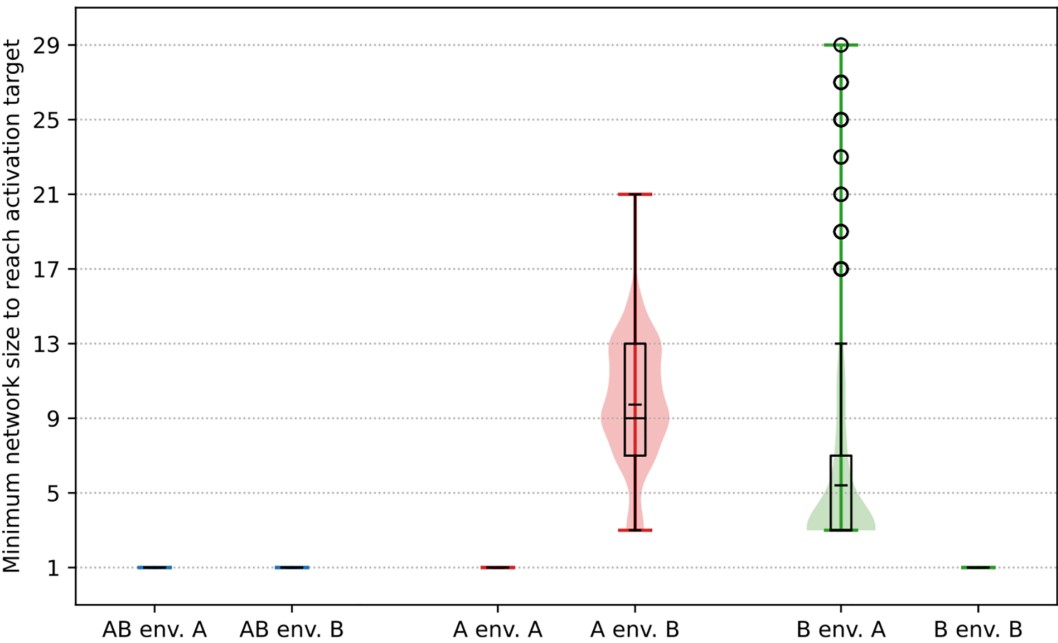

**Fig 8. Minimal contiguous subnetwork size required for the central gene in the subnetwork to present the same activation state as in the complete genome, for each gene type, and in each environment.** The data is computed only for genes in the best final individual of each replicate which present the correct activation state in both environments (97.7% of *AB* genes, 92.7% of *B* genes and 53.2% of *A* genes). In each case, a box plot is overlaid on a violin plot representing the whole distribution, the mean is represented by the smaller tick in the box plot, and circles denote outliers.

activation in the small subnetwork to inhibition in the large subnetwork, and in the two subnetworks centered around the *A* gene (bottom), the two *B* genes and two out of the three central *A* genes also switch from activation to inhibition when moving from the small to the large subnetwork. In both examples, the activity of a gene does therefore not only depend only on its interaction with its closest neighbors, but with a broader section of the genome.

We then computed these minimal subnetwork sizes for every gene that presents the correct activation state in each environment in the final best individual of each of the 30 replicates. Markedly different patterns again appear, depending on whether the targeted behavior for the gene is activation or inhibition, as depicted in Fig 8. For *AB* genes in both environments, as well as for *A* genes in environment A and *B* genes in environment B, the experimentally obtained minimum subnetwork size is 1, consistently with the supercoiling response of an isolated gene (previously shown in Fig 4). Indeed, with a basal supercoiling value of $\sigma_{basal}$ = −0.06, an isolated gene experiences a high expression level in both environments–even without interactions–in the model, as in real-world bacteria.

When the expression target of the gene is inhibition, that is for *A* genes in environment B and for *B* genes in environment A, the picture is however quite different. In this case, a significantly larger subnetwork is required in order to obtain inhibition of the focal gene: The median subnetwork size required to inhibit *A* genes in environment B is 9, or 4 genes on each side. For *B* genes, the median subnetwork size for inhibition in environment A is lower than that required for the inhibition of *A* genes in environment B, but higher than when the target is activation: Genes always need at least a subnetwork of size 3 (1 gene on each side), and several outliers need a subnetwork of more than 20 genes in order to obtain inhibition.

The gene interaction networks that evolve through the transcription-supercoiling coupling therefore exhibit a structure that cannot be explained only by local interactions, but that can on the contrary require the participation of a significant number of genes in order to allow genes to reach their required environment-specific expression levels.

## 2.4. A whole-genome gene interaction network

Having shown that the transcription-supercoiling coupling plays a major role in the regulation of gene expression in our model, and that supercoiling-mediated interactions can implicate more than just neighboring genes, we then sought to describe these interactions in more detail by viewing the genome as a network of interacting genes. The matrix of gene interactions used to compute gene expression levels, whose coefficients $\frac{\partial \sigma_i}{\partial e_j}$ represent the effect of the transcription of every gene on the local level of supercoiling at every other gene (and decrease linearly with distance, see Equation 1 in Methods), could seem to provide a natural graph representation of the interactions between the genes in the genome of an individual. However, as this matrix does not take into account actual gene expression levels, using it directly could provide an inaccurate picture of the effective interactions between genes (for example, overestimating the influence of a weakly-expressed gene). We therefore constructed an *effective* interaction graph, by measuring instead the effect of gene knockouts on gene expression levels.

**Gene knockouts.** Gene knockout is a genetic technique in which a gene of interest is inactivated (knocked-out) in order to study its function (in our case, its possible role as part of a gene interaction network). In order to knock out a given gene in an individual in our model, we set the transcription rate of that gene to zero during the computation of gene expression levels (as described in Methods). This mimics a loss of function of the promoter of the gene, while keeping the intergenic distance between its upstream and downstream neighbors unchanged, thereby minimizing differences to the original individual. The result of a gene knockout on the genome of an evolved individual is shown in Fig 9. The knocked-out gene is gene 36 (bottom left of the genome), which is of type *AB* and originally activated in both environments (see Fig 1 for the original genome of the same individual). We can see that, in environment A, knocking out this gene results in a switch in the activation state of 7 genes (hatched in the left-hand side of Fig 9), and that these genes are not all contiguously located. This knockout also results in local supercoiling changes that propagate up to the bottom-left third of the genome, outside of the direct influence of the knocked-out gene. In environment B, knocking out this gene instead results in milder supercoiling changes that do not result in any gene switching state. In this example, knocking out even a single gene can therefore substantially affect gene expression levels, and lead to a switch in the activation state of other genes on the genome, even when these genes do not directly interact with the knocked-out gene.

**Constructing the effective interaction graph.** We construct the effecting interaction graph in the following manner: we successively knock out every gene in the genome, and each time add edges from the knocked-out gene to every other gene whose activation state is switched by the knockout in either environment. If the knockout switches off a gene that was originally activated in the complete genome, we mark the edge as an activation edge, meaning that the knocked-nout gene was necessary in order to activate the switched-off gene. If the knockout conversely switches on a gene that was originally inhibited in the complete genome, we mark the edge as an inhibition edge. If knocking out a gene switches on or off the same other gene in the two environments, we only add a single edge (even if one edge is an activation edge and the other an inhibition edge), as our main focus is on the connectedness of

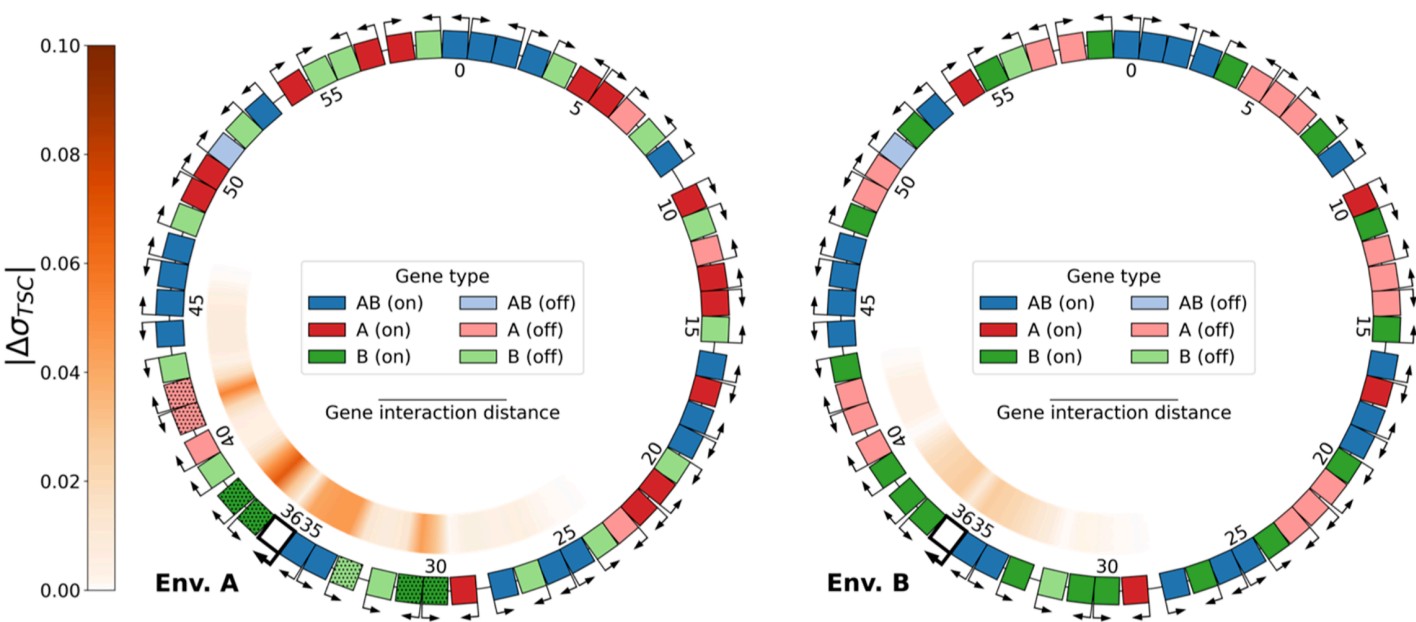

**Fig 9. Knockout of gene 36 (of type *AB*, in bold, colored white) of the final best individual of replicate 21, evaluated in environments A (left) and B (right).** Hatched genes represent genes whose activation state is switched by the knockout when compared to the original genome. The inner ring represents the absolute difference $|\Delta\sigma_{TSC}|$ in the level of transcription-generated supercoiling between the knockout genome and the original genome (shown above in Fig 1).

the resulting graph. The effective interaction graph of the example evolved individual in Fig 9 is presented on the left-hand side of Fig 10. In this individual, there is only one weakly connected component (WCC), meaning that all the genes of this individual contribute to a single, whole-genome interaction network.

**Global structure of the effective interaction graphs.** In order to characterize the effective interaction graphs of evolved individuals, we compared them with the effective interaction graphs of 30 random individuals drawn using the same genome parameters (shown in Table 2 in Methods) as the initial individuals used at the beginning of evolution. The distribution of WCC sizes for each group of graphs are presented on the right-hand side of Fig 10. As we can see, the effective interaction graphs of evolved individuals are clearly different from those of random individuals. Indeed, evolved genomes have WCC sizes of 58 to 60 genes (left), comprising every or nearly every gene on the genome, along with very few single-gene WCCs. In particular, in 26 out of the 30 evolved populations, the interaction graph of the best individual comprises only a single WCC that includes every gene on the genome, similarly to the interaction graph in Fig 10. In random genomes (right), on the contrary, WCC sizes span the whole range from single-gene to whole-genome WCCs, with most of the connected components counting less than 10 genes.

**Local structure of the effective interaction graphs.** Evolved genomes are indeed on average much more connected than random genomes, as we can see in Fig 11, which presents the out- and in-degree of genes (averaged by gene type) in the effective interaction graphs. The left-hand side of Fig 11 first shows the average out-degree of each gene type (i.e., the number of genes that are switched either on or off by knocking out a gene of that type). While knocking out a gene in a random genome switches the state of a little less than 2 other genes on average, independently of the type of the knocked-out gene, this number is much higher in

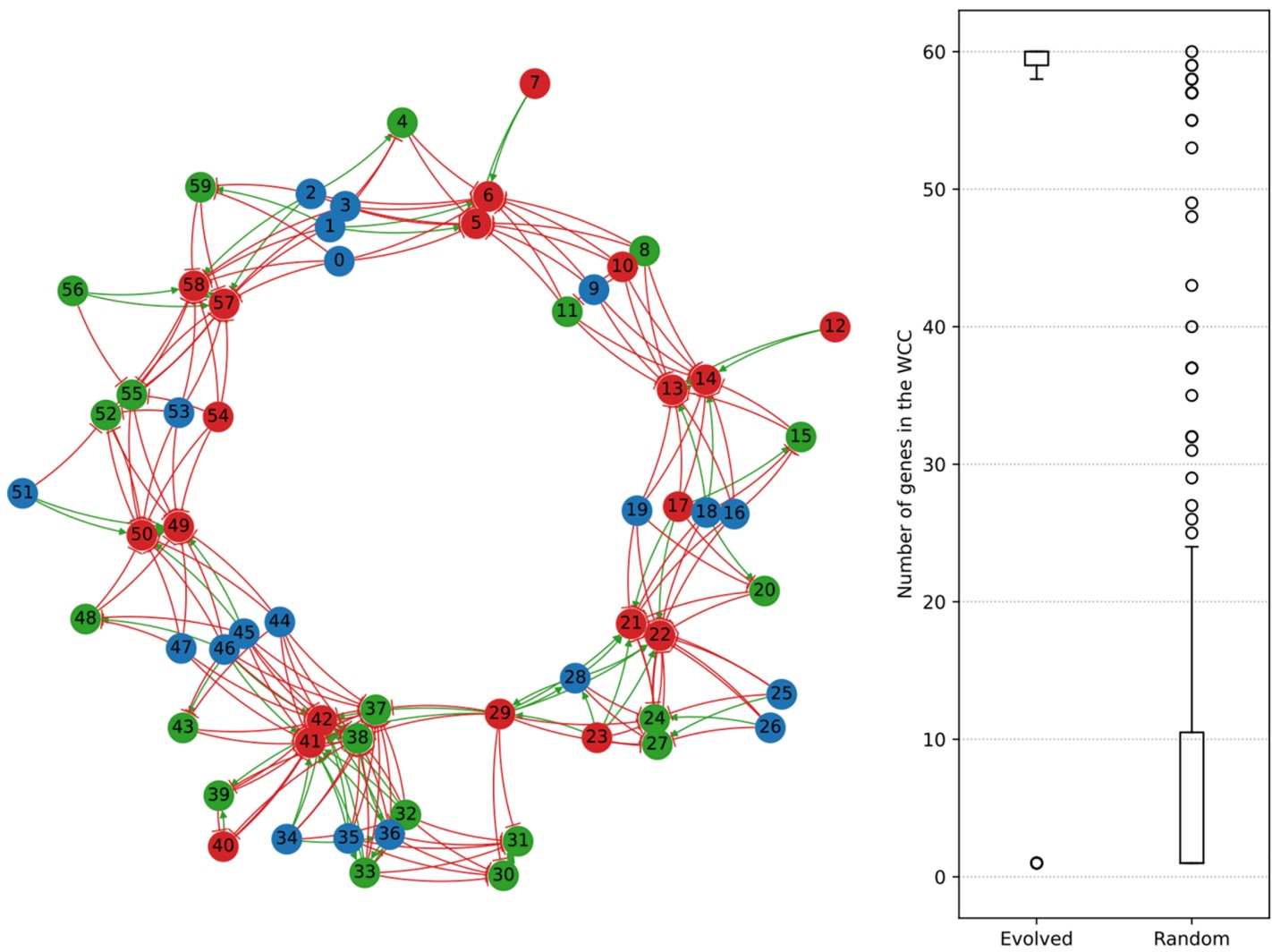

**Fig 10. Left: effective interaction graph of the best individual at the last generation of replicate 21, obtained by knocking out every gene one by one and measuring the resulting gene switches in either environment.** Activation edges are drawn in green, and inhibition edges in red. Gene numbering is the same as in Figs 1, 8 and 9. Right: box plot representing the distribution of weakly connected component (WCC) sizes in the effective interaction graphs of evolved individuals (left) compared to random individuals (right). Circles outside the box plot whiskers denote outliers.

evolved genomes. Knocking out *A* or *B* genes switches 4 other genes on average, and knocking out *AB* genes switches up to 7 other genes. Through this higher connectedness, *AB* genes therefore play a quantitatively more important regulatory role than *A* genes or *B* genes. This can be explained by the fact that *AB* genes are activated–and generate more supercoiling through transcription, as shown in Fig 5–in both environments, while most *A* and *B* genes are instead inhibited in one environment or the other.

When looking at the in-degree of genes (the number of genes whose knockout will switch a given gene on or off) on the right hand-side of Fig 11, we can see that evolved genomes are again much more connected on average than random genomes, and that the in-degree of genes greatly depends on their type. Indeed, *AB* genes are only switched by one other gene on average, meaning that their activation state is robust to perturbations in the interaction

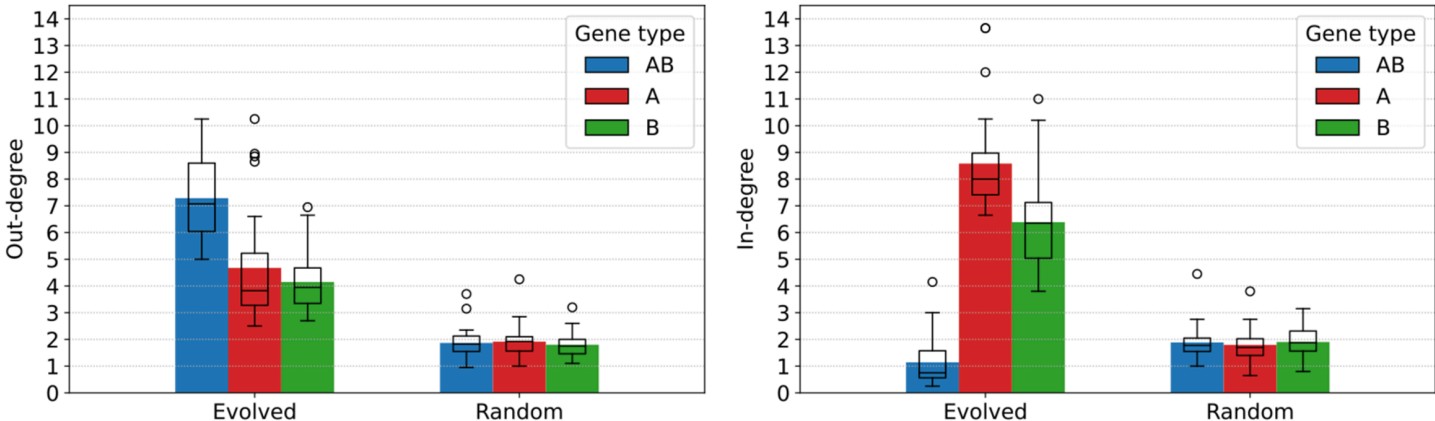

**Fig 11. Left: average out-degree (number of genes switched by knocking out a given gene) of the nodes in the effective interaction graph, separated by gene type, for evolved and random individuals.** Right: average in-degree (number of genes whose knockout switches a given gene) of the nodes in the effective interaction graph, separated by gene type, for evolved and random individuals. Box plots show the median and dispersion around the average value, and circles represent outliers.

network. The robustness of *AB* genes could be expected, as these genes must have the same activation state in both environments in order to attain high fitness. *A* genes and *B* genes have an oppositely much higher in-degree, meaning that their activation state relies on the regulatory action of a large number of other genes. Similarly to *AB* genes, this high connectedness could be expected, as a high in-degree could make *A* and *B* genes more sensitive to variations between the two environments.

In our model, the evolution of relative gene positions on the genome therefore integrates local supercoiling-mediated interactions between neighboring genes into a single genome-wide interaction network, through which all genes interact in order to reach their targeted expression levels.

## 3. Discussion and perspectives

DNA supercoiling, through its effect on promoter activation and hence on gene transcription [4], is an important actor of the regulatory response of bacteria to diverse environmental conditions [9]. In return, gene transcription impacts the level of DNA supercoiling, through a mechanism first described in the twin-domain model of supercoiling [10], and plays a major role in shaping the bacterial DNA supercoiling landscape [13], through what has been termed the transcription-supercoiling coupling [27,29]. Taken together, these observations suggest that supercoiling-mediated interactions between the transcription rates of neighboring genes could play a part in regulating bacterial gene activity, and might hence influence the evolution of bacterial genome organization. A growing body of experimental assays is nowadays increasingly lending support to this hypothesis (reviewed in Table 1). However, to the best of our knowledge, no study has until now explored the impact of the transcription-supercoiling coupling at the genomic scale, either from an experimental or computational perspective.

In this work, we thus sought to assess the possibility of the evolution of such supercoiling-based gene regulation in an *in silico* bacteria-like model, and to determine potential hallmarks on the local and global organization of bacterial genomes of this mode of regulation. To this end, we developed an evolutionary model of the transcription-supercoiling coupling, in which populations of individuals must evolve differentiated gene expression levels in response to two different environmental conditions. We showed that, in this model, gene regulation by

**Table 1. Recent experimental results showing that genome organization can modulate gene expression through the transcription-supercoiling coupling.**

| Experimental method | Result and link to the TSC | Reference |
|---|---|---|
| Measure gene expression in a 2-gene construct with varying relative orientations, isolated from the rest of the *E. coli* chromosome. | Adding a gyrase binding site between the two genes increases transcription, showing that transcription affects and is affected by local supercoiling. | [35] |
| Measure gene expression in a 2-gene construct, in which genes are co-oriented and located on a plasmid in *E. coli*, with varying promoter sequences. | When the expression of the upstream gene is increased by changing promoter sequence, expression of the downstream gene is decreased, consistently with the twin-domain model. | [44] |
| Sequence diverse *S. enterica* strains and measure differential gene expression between strains. | Within a large-scale chromosomal inversion, differential expression is stronger for genes close to the inversion boundaries. | [38] |
| Measure *yfp* fluorescence under the control of a 3-gene synthetic network inserted in *trans* on the *E. coli* chromosome. | When the *yfp* repressor is in a divergent or upstream position to the other genes, it is more active and fluorescence is lower. | [49] |
| Measure the gene expression response to environment-triggered DNA relaxation in *E. coli* and *D. dadantii*. | Convergent genes are more strongly activated than divergent genes by DNA relaxation. | [17] |
| Measure the gene expression response to multiple environmental perturbations in *M. pneumoniae*. | Gene expression is globally anticorrelated for convergent genes. | [22] |
| Compare gene expression by relative pair orientation between wild-type and evolved genomes after 20,000 generations in the LTEE. | As the genome evolved to be more negatively supercoiled, convergent genes saw a lower increase in expression than divergent genes, consistently with the twin-domain model. | [17] |
| Measure gene expression in a 2-gene inducible construct with varying relative orientations on a plasmid in *E. coli*. | Gene expression reacts differently to the induction depending on relative orientation. | [34] |
| Insert a cassette with divergent genes on the *E. coli* chromosome and measure expression of one gene as the other is induced. | An increase in expression of the induced gene is matched by an increase in expression of the divergent neighbor. | [37] |

DNA supercoiling is a sufficient mechanism to evolve environment- and gene- specific patterns of activation and inhibition. In particular, we observed the emergence of relaxation-activated genes that respond to supercoiling oppositely to the majority of genes, which are classically inhibited by DNA relaxation [46]. Our results therefore demonstrate that, in theory, this response to supercoiling can result not only from promoter sequence as in the case of the relaxation-activated *gyrA* promoter [47] or spacer length [48], but also from genome organization in itself, as had already been suggested by [37] or [17]. As such, these findings suggest that supercoiling-mediated regulation could be a sufficient mechanism to adapt gene expression in response to environmental constraints. This regulatory role could in particular be especially important in bacteria that lack other regulatory mechanisms, such as the genome-reduced *B. aphidicola* [21] or *M. pneumoniae* [22].

Having shown that the transcription-supercoiling can be leveraged to tune gene expression levels in bacteria, we investigated the patterns of genome organization that underlie this transcriptional response to different supercoiling environments. At the most local scale, we found that evolved genomes are enriched in divergent pairs of always-on genes that form positive feedback loops, as well as in convergent pairs that oppositely act as bistable toggle switches controlled by supercoiling rather than by transcription factors [45]. The existence of such supercoiling-mediated toggle switches had been earlier posited by using mechanistic biophysical models that explicitly describe the movement of RNA polymerases during gene transcription [42,43], and their emergence in our model suggests that such toggle switches could indeed evolve as a means to regulate the expression of neighboring genes. Note that

other models, such as [33], paint a more complex picture and suggest that divergently transcribing polymerases could slow down transcription, leading to a more complex interaction between neighboring genes, such as that observed in [35]. We additionally observed that, similarly to pairs, the distribution of triplets of neighboring genes in evolved genomes is not uniform. Although the distribution of triplet frequencies partly reflects the non-uniformity of the pairs that underlie these triplets (as discussed in the Results), this observation may reflect more complex favorable three-way interactions such as external input leading to the activation of one or the other gene in a toggle switch–see a systematic exploration in [49]. Finally, we showed that these local interactions between pairs or triplets are in fact not entirely sufficient to selectively activate or inhibit genes in specific environments, but that interactions between larger groups of genes can be required to do so. Such regulation of gene expression through the interaction of groups of co-located genes could help explain the persistence of synteny segments (clusters of genes that display correlated transcription levels at the supra-operonic scale) that has been evidenced in bacterial evolutionary histories [40]. Indeed, if local interactions play a role in regulating the expression of groups of neighboring genes, genomic rearrangements that alter relative gene positions within these structures could disrupt their regulation and hence be evolutionarily unfavorable. Finally, we characterized in further detail the gene interaction networks that evolve in the model by adapting the classical genetics tool of gene knockouts [50]. We showed that supercoiling-mediated interactions integrate the entire genome of evolved individuals into a single connected network of interacting genes, in opposition to the sparse, disconnected networks displayed by randomly generated individuals. Moreover, we showed that genes play different roles in these networks depending on their type, corresponding to the type-specific responses to environmental variations that genes must display in the model. Overall, our simulations therefore demonstrate that, in this model, the transcription-supercoiling coupling provides a strong and precise regulatory mechanism that allows for the evolution of complex regulation patterns based solely on the relative positions of genes on the genome, and that this regulation can be sensitive to even very small environmental perturbations (see Figs A-D in S1 Text). Finally, our simulations show that this particular mode of regulation could impact the structure of bacterial genomes not only at a local scale, but also at a wider scale through supercoiling-mediated regulatory networks.

In this work, we voluntarily kept our model as simple as possible in order to obtain easily interpretable results, while retaining the core concept of the transcription-supercoiling coupling. In particular, in order to keep simulations computationally tractable, we used small genome and population sizes compared to real bacteria, but we do not expect these modeling choices to impact our results (see Methods). In further work, an number of adjacent relevant questions could also be studied by introducing other elements to the model. For example, we chose to use a deterministic algorithm to compute the gene expression levels resulting from the transcription-supercoiling coupling for a given value of $\delta\sigma_{env}$, but stochastic approaches could be considered, such as averaging final expression levels over slightly perturbed initial gene expression levels. We do not expect such model changes to qualitatively affect the results, but they could provide additional information about the stability of the evolved gene regulatory networks. Increasing the number of environments that individuals face, and accordingly increasing the number of gene types, might provide a larger panel of evolutionarily selected local gene organizations that might reflect more accurately the real-world diversity of bacterial genomes. In another direction, modeling more explicitly polymerases and topoisomerases would allow us to study how including transcriptional read-through (the transcription of successive genes by a single RNA polymerase) and topological barriers to supercoiling would alter the regulatory genomic structures that evolve in our model. Indeed, this mechanism

has been hypothesized to play a part in the evolutionary conservation of synteny segments in bacterial genomes, by correlating the expression levels of genes in these segments [40]. Similarly, letting the response to supercoiling of gene promoters coevolve with genomic organization could help understand the evolution of unusual promoters such as the *gyrA* promoter, which is activated by relaxation due to its particular sequence [47]. Finally, integrating a classical model of gene regulation via transcription factors to our model, such as the one presented by [51], could also help shed light on the coevolution between the different modes of gene regulation that are available to bacterial genomes. From a theoretical standpoint, a range of mechanistic biophysical models of the transcription-supercoiling coupling have been put forward, using different hypotheses in order to address related questions on this topic. [52] show a phase transition in the transcription regime as the number of RNA polymerases transcribing a given gene increases; [42] show that bursty transcription can emerge from the transcription-supercoiling coupling; and [29] and [17] try to predict gene expression levels quantitatively, as a function of the local level of DNA supercoiling. An important validation of these complementary approaches would therefore be to investigate the extent to which these models, including ours, conform to one another as the level of abstraction changes. From an experimental standpoint, the advent of long-read DNA sequencing now allows for the study of genome-scale structural variations in multiple strains of the same species, and such data could be used to systematically study the link between genome organization and gene expression, furthering the work of [39] or [38]. However, such an analysis remains out of the scope of the current paper. Finally, from a synthetic biology point of view, a better understanding of the regulatory interactions that result from the transcription-supercoiling coupling could help design more finely controlled artificial genetic constructs [35,43,44].

## 4. Conclusion

To the best of our knowledge, this work is the first to propose a model to investigate the role of the coupling between gene transcription and DNA supercoiling in the evolution of the structure of bacterial genomes. By integrating a biophysical model of the transcription-supercoiling coupling into an evolutionary simulation, we have demonstrated the theoretical possibility of the evolution of gene regulation through a supercoiling-mediated regulatory network that allows for precise responses to variations in the environment. For experimentalists, this work builds up on a series of theoretical models that could help explain the heterogeneous transcriptomic response (with both up- and down-regulation of multiple genes) observed in bacteria confronted to supercoiling variations, due for example to virulence-inducing environments [53] or to gyrase-inhibiting antibiotics [54]. It also suggests further experiments that should help better understand the interaction between the transcription-supercoiling coupling and bacterial genome organization. For evolutionists, this work shows that genomic inversions can provide an unforeseen source of adaptive mutations through supercoiling-mediated regulatory rewiring at their boundaries, explaining fitness gains observed in strains harboring genomic inversions [38,39]. Overall, it therefore lends further weight to the hypothesis that supercoiling-based regulation could help conserve local gene order throughout evolutionary histories [40]. Finally, for synthetic biologists, it provides a theory that could help predict more accurately the transcription levels that can be expected to result from a given gene context [44], and could thus help in the design of new artificial genetic circuits.

## 5. Methods

This section details the model that we use throughout the manuscript to study the role of the transcription-supercoiling coupling in the evolution of gene regulation and genome organization in bacteria-like organisms. The model consists in an individual-based evolutionary simulation, in which individual whose phenotypes are computed according to a biophysical model of the effect of supercoiling on gene expression must adapt to two environments presenting different supercoiling levels. It is based upon and refines our previous model presented in [55]. We start by presenting the individual-level biophysical model, and describe how we compute gene expression levels based on their relative positions on the genome by taking into account the transcription-supercoiling coupling. Then, we describe how we compute the fitness of individuals by evaluating their adaptation to the two environments, and describe how we create new individuals and populations with genomic inversions. Finally, we present the experimental setup that we used in order to run the simulations presented in the Results section, and discuss code and data availability.

### 5.1. Individual-level model of the transcription-supercoiling coupling

We define the genotype of an individual as a single circular chromosome, meant to represent a bacterial chromosome. The chromosome consists in a fixed number of protein-coding genes, which are separated by non-coding intergenic segments of varying sizes, and is additionally characterized by a basal supercoiling level $\sigma_{basal}$. Each gene on the chromosome is characterized by its starting position (note that genes cannot overlap in our model), its orientation (on the forward or reverse strand), its length, and its basal expression level. We always consider individuals within an environment, which we define by the perturbation $\delta\sigma_{env}$ that it imposes to the background supercoiling level of the chromosome. We define the phenotype of an individual in a given environment as the vector that holds the expression levels of all of its genes. We compute this phenotype by solving the system of equations given by the interaction of the individual's genes with one another through the transcription-supercoiling coupling (described below), on a chromosome with a background supercoiling level of $\sigma_{basal} + \delta\sigma_{env}$. Note that, while we do not have a formal proof that this system always has a single solution, this is empirically the case in the parameter range used in the simulations.

The genome of an example individual with 20 genes is shown on the outer ring of the left-hand side panel of Fig 12. The inner ring depicts the local level along the genome of DNA supercoiling resulting from gene transcription, when this individual is evaluated in an environment with a supercoiling shift of $\delta\sigma_{env} = 0$. As expected from the twin-domain model of supercoiling, we can observe a buildup in negative supercoiling (in blue) between pairs of genes in divergent orientations, such as the C-D or F-G gene pairs, and a buildup in positive supercoiling (in red) between pairs of genes in convergent orientations, such as the K-J or Q-R gene pairs. The right-hand side panel of Fig 12 shows the computation of the gene expression levels for this individual in the same environment (as detailed below). Note that, in this model and throughout the manuscript, we conflate gene transcription rates with mRNA concentrations, as we assume that mRNAs are degraded at a constant rate, and as transcription rates in our model are only affected by the effect of supercoiling on transcription. We additionally conflate transcription rates with expression levels (or protein concentrations), as we again assume proteins to be translated at a rate proportional to the associated mRNA concentrations and degraded at a constant rate.

**Effect of transcription on supercoiling.** For an individual with a genome containing $n$ genes, we model the influence of the transcription of each gene on the level of supercoiling at

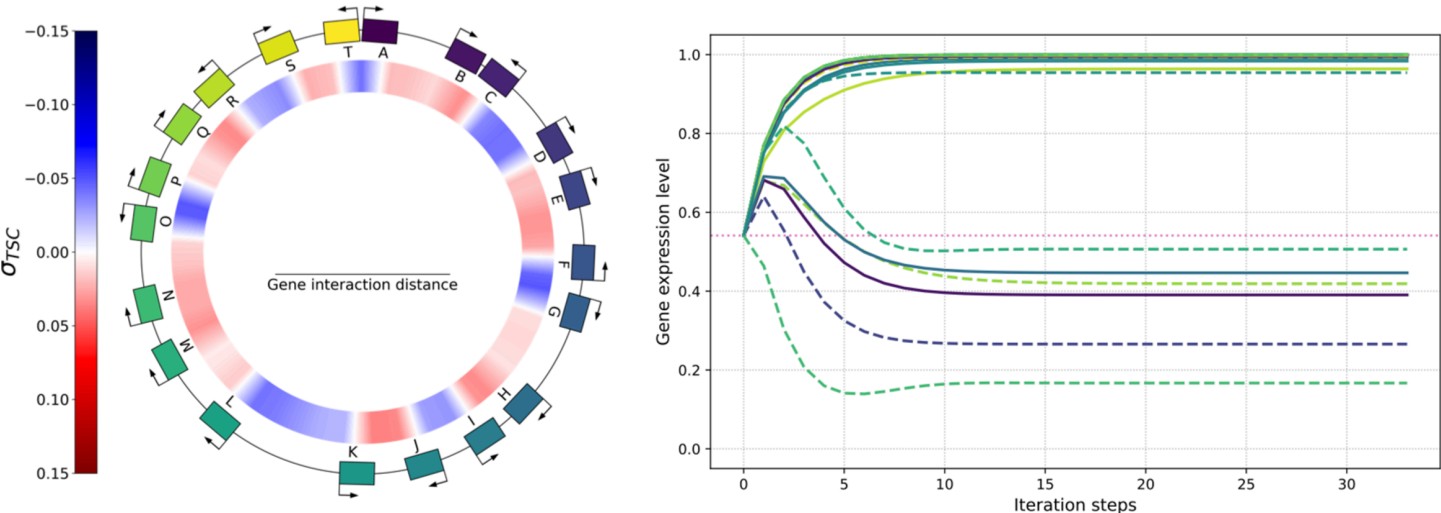

**Fig 12. Left: genome (outer ring) and level of transcription-generated supercoiling ($\sigma_{TSC}$, inner ring) of an example individual with 20 genes placed at random positions and orientations and colored by position, with a gene length and average intergenic distance of 1 kb each, and a basal supercoiling level of $\sigma_{basal} = -0.066$.** The individual is evaluated in an environment in which $\delta\sigma_{env} = 0$. Right: evolution of the expression level of each gene of the individual (reusing gene colors from the genome) during the computation of the solution to the system given by equations 2, 3, and 4, starting from initial expression levels of $e_{1/2}$. Solid lines represent genes on the forward strand, dashed lines genes on the reverse strand, and the dotted pink line represents $e_{1/2}$, the gene activation threshold.

the promoter of every other gene in the form of an *n*-by-*n* matrix, which we call the interaction matrix. The coefficient $\frac{\partial\sigma_i}{\partial e_j}$ at indices (*i*,*j*) in this matrix represents the infinitesimal variation in DNA supercoiling at the promoter of gene *i* due to the transcription of gene *j*. The value of this coefficient is given by Equation 1:

$$\frac{\partial\sigma_i}{\partial e_j} = \eta \cdot c \cdot \max\left(1 - \frac{d(i,j)}{d_{max}}, 0\right) \tag{1}$$

$\eta$ gives the sign of the interaction, which depends on the position and orientation of gene *j* relative to gene *i*, according to the twin-domain model [10]. If gene *j* is upstream of gene *i*, and if it is on the same strand as (points towards) gene *i*, then its transcription generates a buildup in positive supercoiling at gene *i* ($\eta = 1$). Conversely, if gene *j* is upstream of gene *i* but on the other strand than (points away from) gene *i*, it generates a buildup in negative supercoiling at gene *i* ($\eta = -1$). If gene *j* is instead located downstream of gene *i*, the sign of the interaction in each case is reversed: $\eta = -1$ if the genes are on the same strand, and $\eta = 1$ otherwise.

We then apply a torsional drag coefficient *c*, which is a high-level representation of the magnitude of the effect of transcription on the local supercoiling level. Finally, we model this change in supercoiling as linearly decreasing with the distance $d(i,j)$ between genes *i* and *j*. More precisely, we consider the distance between the promoter of gene *i*, the position at which the local level of supercoiling affects the probability that an RNA polymerase binds to the DNA and starts transcribing gene *i*, and the middle of gene *j*, the average location of the RNA polymerases that transcribe gene *j*, assuming that DNA is transcribed at a constant speed. When this distance reaches a threshold of $d_{max}$, we consider the two genes to lie too far away to interact, and the effect vanishes. In other words, $d_{max}$ represents the maximum gene interaction distance on either side of a gene (see Fig 12). Finally, we consider that genes do not interact with themselves through supercoiling, so we set $\frac{\partial\sigma_i}{\partial e_i}$ to 0 for all *i*.

**Effect of supercoiling on transcription.** In order to describe the effect of supercoiling on transcription, we adapted the equations and parameter values presented in [17], which are based on the *in vitro* analysis of the transcription of model bacterial promoters, for which expression has been shown to increase sigmoidally with negative supercoiling [46,48]. We first compute the local level of supercoiling $\sigma_i$ at the promoter of gene $i$, which is the sum of the background supercoiling level $\sigma_{basal} + \delta\sigma_{env}$ (which is constant along the genome for any given individual in a given environment), and of the local variation in supercoiling caused by the transcription of every other gene (represented in Fig 12 as $\sigma_{TSC}$):

$$\sigma_i = \sigma_{basal} + \delta\sigma_{env} + \sum_{j=1}^{n} \frac{\partial\sigma_i}{\partial e_j} e_j \tag{2}$$

We then compute the expression level of the gene using a thermodynamic model of transcription, in which transcription is approximated by a sigmoid that increases with negative supercoiling until a saturation threshold is reached. More precisely, we compute the opening free energy $U_i$ of the promoter of gene $i$, which depends on $\sigma_i$, the level of supercoiling at the promoter, according to the following sigmoidal function:

$$U_i = \frac{1}{1 + e^{(\sigma_i - \sigma_{1/2})/\epsilon}} \tag{3}$$

This sigmoid function has two fixed parameters: $\sigma_{1/2}$, the level of supercoiling at which the opening free energy is at half its maximum level, and $\epsilon$, which tunes the sensitivity of promoter opening free energy to supercoiling variations. We can then compute the expression level $e_i$ of gene $i$, using an inverse effective thermal energy $m$:

$$e_i = e^{m(U_i - 1)} \tag{4}$$

The transcription level of a gene is therefore expressed in arbitrary units between $e^{-m}$, the minimum expression level when the promoter is most hindered by supercoiling (when $U_i = 0$), and 1, the maximum expression level, when the promoter is most activated by supercoiling (when $U_i = 1$). Throughout the manuscript, we describe a gene as activated if its transcription level is above the mean of these two values $e_{1/2} = \frac{1}{2}(e^{-m} + 1)$, and inhibited otherwise.

**Computation of gene expression levels.** Recall that we define the phenotype of an individual in an environment (described by $\delta\sigma_{env}$) as the vector of gene expression levels that is solution to the system of equations given by Equations 2, 3 and 4, in that environment. In order to compute this phenotype, we numerically compute a solution to the system of equations by using an iterative fixed-point algorithm, starting from an initial state in which all genes are expressed at $e_{1/2}$. Note that, while there might be several stable or unstable fixed points of the system for a given genome and environment, our algorithm in practice always converges to a fixed point. We therefore define this point as the gene expression levels of that individual in that environment. A representative example of this computation is shown in the right-hand side panel of Fig 12.

## 5.2. Evolutionary model

Equipped with a model of the coupling between DNA supercoiling and gene transcription at the whole-genome scale, we now embed it into an evolutionary framework. More precisely, we model the evolution of a population of individuals, each behaving as described in Sect 5.1, which must simultaneously adapt to two distinct environments named A and B, as measured

by an increase in fitness $f$ (defined below) over time. Environment A induces DNA relaxation, with a supercoiling shift of $\delta\sigma_{env} = \delta\sigma_A = 0.01 > 0$, and environment B induces negative DNA supercoiling, with a supercoiling shift of $\delta\sigma_{env} = \delta\sigma_B = -0.01 < 0$. In order to represent adaptation to these environments, we assign genes in the individual's genomes to three categories, representing different target gene expression levels, in equal proportions: *AB* genes should be expressed in both environments, akin to housekeeping genes; *A* genes should be expressed in environment A but not in environment B; and, conversely, *B* genes should be expressed in environment B but not in environment A, both representing environment-specific genes, such as the pathogenic genes of *S. enterica* or *D. dadantii* [8,56].

**Fitness.** The fitness of an individual is computed using its gene expression levels in each of the two environments. Let $(e_A^A, e_B^A, e_{AB}^A)$ be the 3-dimensional vector representing the average gene expression level per gene type of an individual in environment A, and $(e_A^B, e_B^B, e_{AB}^B)$ be the average gene expression per gene type of this individual in environment B. Let $(\tilde{e}_A^A, \tilde{e}_B^A, \tilde{e}_{AB}^A)$ and $(\tilde{e}_A^B, \tilde{e}_B^B, \tilde{e}_{AB}^B)$ be the target expression values for each gene type in each environment, reflecting the gene type definitions presented above. For environment A, we set $\tilde{e}_A^A = \tilde{e}_{AB}^A = 1$, and $\tilde{e}_B^A = e^{-m}$, which are respectively the maximal and minimal attainable gene expression levels in the model. Similarly, for environment B, we set $\tilde{e}_B^B = \tilde{e}_{AB}^B = 1$, and $\tilde{e}_A^B = e^{-m}$. We can then compute the sum $g$ of the squared error (or gap) between the mean and targeted expression levels for each gene type in each environment:

$$g = \sum_{i \in \{A,B,AB\}} \left(e_i^A - \tilde{e}_i^A\right)^2 + \sum_{i \in \{A,B,AB\}} \left(e_i^B - \tilde{e}_i^B\right)^2 \tag{5}$$

Finally, we define the fitness of the individual as $f = \exp(-k \cdot g)$, where $k$ is a scaling factor representing the intensity of selection: as $k$ increases, the difference in fitness, and hence in reproductive success, between individuals with different values of $g$ also increases.

**Evolutionary algorithm.** We consider populations of $N$ individuals, which reproduce in non-overlapping generations. At each generation, we first compute the fitness of each individual in that generation, based on its gene transcription levels in each environment (as described above). Then, in order to create the following generation, we draw $N$ reproducers at random, proportionally to their fitness, and with replacement (meaning that a high-fitness individual can have several offspring) from the current generation. Finally, we create the offspring of each reproducer by randomly applying mutations to the genome of the reproducer, resulting in the new generation.

**Mutational operator: Genomic inversions.** In order to model the evolution of genome organization, the only mutational operator that we use is genomic inversions. This allows for the reordering of genes on the genome through series of inversions, modeling observations in certain *E. coli* [39] or *S. enterica* strains [38]. We do not include large-scale duplications or deletions, as these rearrangements would possibly change the number of genes. In other words, we assume gene loss or duplication to be lethal mutations in this model. Note that translocations can be modeled as a series of well-chosen consecutive inversions, and are therefore implicitly present in our model.

In order to perform a genomic inversion, we choose a start point and an end point uniformly at random in the non-coding intergenic sections of the genome. This ensures that genes cannot be broken apart by inversions (remember that we assume that gene losses are lethal). Having chosen the ends of the inversion, we extract the DNA segment located between these points and reinsert it at the same position, but in the reverse orientation. The inversion thereby reverses the orientation of every gene inside the segment, but conserves the

relative positions and distances between these genes. The intergenic sections at the boundaries of the inversion can however grow or shrink depending on the position of its start and end points, thereby allowing intergenic distances to change over evolutionary time. Note that the total amount of intergenic material, which is a parameter of the simulation (see next subsection), is itself kept constant by this operation.

Finally, when mutating an individual, we start by drawing the number of inversions to perform from a Poisson law of parameter $\lambda = 2$, meaning that the offspring of an individual will on average undergo two inversions. Then, we perform each inversion in succession as previously described, in order to obtain the final mutated offspring.

### 5.3. Experimental setup

In order to conduct the simulations presented in the Results section, we let 30 independent populations of $N = 100$ individuals evolve for 1,000,000 generations. We initially seeded each population with 100 clones of a randomly generated individual with 60 genes, or 20 genes of each type ($A$, $B$ and $AB$), using a different seed for each population. The parameter values that we used are given in Table 2, and can be broadly grouped into genome-level parameters (gene length, intergenic distance, basal supercoiling level and supercoiling transmission distance) and promoter-level parameters (promoter opening threshold and effective thermal energy, crossover width). Both the genome-level parameters that describe the chromosome and the promoter-level parameters used to compute the transcriptional response to supercoiling were taken from averaged experimental values measured in *E. coli*. Note that the supercoiling transmission distance $d_{max}$ is expected to be heterogeneous along a real chromosome; in our experiments, choosing larger values of $d_{max}$ did not lead to qualitatively different results, but values smaller than 4 kb resulted in the loss of inhibition of genes *A* in environment B (see Figs E and F in S1 Text). Also note that the torsional drag coefficient $c$ is a new parameter that we introduce in this model to represent the influence of torsional drag on the local level of supercoiling. We have empirically chosen its value so that this effect is of the same magnitude as the other sources of supercoiling variations (i.e., environmental perturbations) in the model.

### 5.4. Genome size, population size, and mutation rate

In order to keep simulations computationally tractable, we had to make trade-offs when choosing parameter values for the size of individual genomes, the size of the population, and the mutation rate. First, we restricted the number of genes in each individual to 60 (20 of each type). While this is much fewer than the around 4,300 genes in the *E. coli* genome [36], our

Table 2. **Parameter values of the transcription-supercoiling coupling model used in the evolutionary simulations.** The upper set of parameters is the genome-level parameters, the lower set the promoter-level parameters, both taken from the *E. coli* literature; the middle parameter is a new addition from our model.

| Parameter | Symbol | Value | Reference |
|---|---|---|---|
| Gene length | $l$ | 1,000 bp | [36] |
| Initial intergenic distance | $d_0$ | 125 bp | [36] |
| Supercoiling transmission distance | $d_{max}$ | 5,000 bp | [14] |
| Basal supercoiling level | $\sigma_{basal}$ | −0.066 | [25] |
| Torsional drag coefficient | $c$ | 0.03 | |
| Promoter opening threshold | $\sigma_{1/2}$ | −0.042 | [17] |
| Supercoiling sensitivity of promoter opening | $\epsilon$ | 0.005 | [17] |
| Inverse effective thermal energy | $m$ | 2.5 | [17] |

model importantly keeps chromosome size much larger than the supercoiling interaction distance, so that each gene can only interact directly with a small proportion of the genome. An increase in the number of genes in the model should therefore not affect the local- and medium-scale patterns that we observe, nor the formation of large-scale interaction networks (although such genomes might possibly harbor several large weakly connected components rather than a single one).

We also chose a population size $N$ much smaller, and a mutation rate $\lambda$ much higher, than encountered in real bacterial populations. This *rescaling* approach is common in computational studies, and preserves the value of $N\lambda$ (the expected number of mutations, or genetic diversity, per generation) while allowing for faster simulations, as runtime is approximatively proportional to $N$ only. To check empirically that this choice does not affect our results, we ran additional simulations with constant $N\lambda$, but increasing $N$ (and decreasing $\lambda$) by a factor of 10 and of 100 respectively. As can be seen in Figs G and H in S1 Text, fitness evolves qualitatively similarly in the three setups, as well as differentiated gene expression levels by environment for each gene type, empirically validating our decision to run the main experiment with a small population size. While fitness and gene activation levels can be seen to evolve more slowly per generation in simulations with larger $N$, this could be due to two non-linearities in the rescaling. First, favorable mutations take longer to fix on average in larger populations [57]. Second, rare mutational events that require two inversions on the same genome happen with a frequency proportional to $\lambda^2$. These events could play an important role in the evolutionary process by providing a mechanism for the escape of local fitness peaks, but rescaling by increasing population size and decreasing mutation rate proportionally actually makes these events occur less frequently.

## 5.5. Supplementary simulations

For the simulations presented in Figs A-F in S1 Text, we let 15 additional independent populations evolve for 250,000 generations, for each set of environmental perturbation values ($\sigma_A = 0.001$ and $\sigma_B = -0.001$, and $\sigma_A = 0.0001$ and $\sigma_B = -0.0001$ respectively). For the simulations presented in Figs G and H in S1 Text, we let 10 populations evolve for 100,000 generations only, due to the computational cost of simulating populations with 1,000 and 10,000 individuals.

## 5.6. Reproducibility and data availability

We implemented the simulation in Python, and optimized the computationally heavy parts using the `numba` package [58]. The source code for the simulation, as well as the notebooks used for data analysis, are available online at the following address: https://www.github.com/tgrohens/evotsc and on the Software Heritage archive at the following address: https://archive.softwareheritage.org/swh:1:dir:dc7169fdc35871aa456a7fecd095f7e0758bc368.

Running the complete set of simulations took around 36 hours of computation on a server using a 24-core Intel Xeon E5-2620 v3 @ 2.40GHz CPU, with each replicate running on a single core and using approximately 300 MB of RAM. The data from the main run of the experiment is available online on the Zenodo platform, at the following address: https://doi.org/10.5281/zenodo.7062757. The supplementary data is available at the following address: https://doi.org/10.5281/zenodo.17077963.

## Supporting information

**S1 Text. S1 Text contains Figs A–H, depicting additional simulation results.**
(PDF)

## Acknowledgments

We would like to thank members of the BioTiC (formerly Beagle) team for insightful conversations, especially Leo Trujillo for his advice on the title of the manuscript.

## Author contributions

**Conceptualization:** Théotime Grohens, Sam Meyer, Guillaume Beslon.

**Formal analysis:** Théotime Grohens, Sam Meyer, Guillaume Beslon.

**Investigation:** Théotime Grohens.

**Methodology:** Théotime Grohens, Guillaume Beslon.

**Software:** Théotime Grohens.

**Supervision:** Guillaume Beslon.

**Validation:** Théotime Grohens, Sam Meyer, Guillaume Beslon.

**Visualization:** Théotime Grohens.

**Writing – original draft:** Théotime Grohens.

**Writing – review & editing:** Théotime Grohens, Sam Meyer, Guillaume Beslon.

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
