## [Decision Letter · Decision Letter 0]

7 May 2025

PCOMPBIOL-D-24-02197

Emergence of Supercoiling-Mediated Regulatory Networks through the Evolution of Bacterial Chromosome Organization

PLOS Computational Biology

Dear Dr. Beslon,

Thank you for submitting your manuscript to PLOS Computational Biology. After careful consideration, we feel that it has merit but does not fully meet PLOS Computational Biology's publication criteria as it currently stands. Therefore, we invite you to submit a revised version of the manuscript that addresses the points raised during the review process.

Please submit your revised manuscript within 60 days Jul 04 2025 11:59PM. If you will need more time than this to complete your revisions, please reply to this message or contact the journal office at ploscompbiol@plos.org. Please include the following items when submitting your revised manuscript:

We look forward to receiving your revised manuscript.

Kind regards,

Tommy Tsan-Yuk Lam, Ph.D.

Academic Editor

PLOS Computational Biology

Stacey Finley

Section Editor

PLOS Computational Biology

**Journal Requirements:**

At this stage, the following Authors/Authors require contributions: Théotime Grohens, and Guillaume Beslon. Please ensure that the full contributions of each author are acknowledged in the "Add/Edit/Remove Authors" section of our submission form.

3) We noticed that you used the phrase 'data not shown' in the manuscript. We do not allow these references, as the PLOS data access policy requires that all data be either published with the manuscript or made available in a publicly accessible database. Please amend the supplementary material to include the referenced data or remove the references.

5) We have noticed that you have uploaded Supporting Information files, but you have not included a list of legends. Please add a full list of legends for your Supporting Information files after the references list.

**Reviewers' comments:**

Reviewer's Responses to Questions

Reviewer #1: This study presents a computational model to analyze the role of DNA supercoiling in bacterial gene regulation and genome evolution. The manuscript is well-written and provides a thorough exploration of the transcription-supercoiling coupling and its implications for bacterial genome evolution. The use of computational modeling to study these interactions is a valuable contribution to the field, and the results potentially provide intriguing insights into the potential evolutionary consequences of supercoiling-mediated regulation.

A major issue with this paper is the lack of analysis on the relationship between the simulation results of proposed model and gene expression dynamics in actual cells. In bacterial cells, gene expression is regulated by transcription factors and other mechanisms. Additionally, during genome evolution, genes that benefit from correlated expression changes tend to form operons through structural mutations. While gene expression regulation via DNA supercoiling undoubtedly exists, its relative contribution compared to other regulatory mechanisms remains unclear. Therefore, it is difficult to assess how much the evolutionary properties identified in this study contribute to fitness in actual evolutionary processes when the simulation considers only DNA-supercoiling-based regulation.

The use of a simplified model incorporating minimal elements to analyze the relationship between gene expression and genome DNA structure in this study is understandable. However, for such a simplified model to be valuable, it should reveal novel insights into biological systems, even if it does not provide precise quantitative correspondence with real data. Unfortunately, the numerical experiments in this study only produce properties that are expected to emerge from the model’s assumptions, i.e., the transcription-supercoiling coupling, making it difficult to recognize novelty in these findings.

To demonstrate the value of this study, the paper needs to establish some form of correspondence with experimental data. The most straightforward simulation result that could be compared with real data would be the relationship between gene orientation and changes in expression levels. For example, in E. coli, transcriptome data from various environmental conditions are available (Sastry, Nature Commun 10, 5536, 2019). If a correspondence could be found between expression changes, gene orientation, and the predicted genome structure as supercoiling in E. coli, the study would gain significant value. Given the abundance and accessibility of E. coli data, it would be a strong candidate for such validation. Another potential approach would be to analyze gene orientation and genome structure in organisms such as Mycoplasma, where regulatory factors are absent.

Reviewer #2: This interesting manuscript focuses on the evolution of genome coiling and how gene transcription is affected by genome coiling, which in term can affect genome fitness. The model is well presented and reasonable to follow, though somewhat suffers from odd choices in model parameters such as tiny population size and massive mutation kernel. Overall the model is well explored and thoroughly investigated for network size, knockouts, and global structure. While there are minor comments below, the major comment is that the biological relevancy appears low- there are a few bacteria that have evolved increased genome supercoiling discussed in the introduction, but even in these examples, protein changes, not inversion drove this genome evolution. It is unclear is any example exists of a gene inversion clearly increasing or decreasing fitness simply through expression effects on super coiling. As such, the mechanism of selection and evolution, while an interesting thought experiment, may have such weak selection coefficients that it is not observed. Is there an example of any gene, in any organism, where an inversion has a measurable effect in any environment on fitness?

Minor comments:

1. Figure 1 would benefit from showing a random individual at the beginning of the experiment, as well as gene transcription levels as another ring.

2. The authors should explain their choice of a population size of 100 individuals and λ=2 inversions per replication event, as well as how their results would be different if population size of much larger and a much smaller mutation kernel were used instead. Given the regime for random selection into evolutionary dynamics, if population sizes of bacteria were huge and selection coefficients of rare inversions were smaller, would the same results be observed?

3. Figures 2 and 3- ‘the first and last decile of the data’s unusual to display and interpret. Why not display a standard deviation?

4. Figure 4 is awesome.

Reviewer #3: In this manuscript, the authors present an analysis coupling a model for DNA supercoiling effects on bacterial gene transcription to evolutionary simulations to investigate whether supercoiling effects and the associated transcriptional effects can impact the organization of bacterial genomes adapting to two different environments with different global supercoiling levels and gene expression requirements, under the assumption that only supercoiling affects transcription (i.e. there are no transcription factors) and using one mutational operator: gene inversion. Their results show that the artificial bacterial genomes can evolve a genome organization in which genes are ordered and oriented so that their expression levels in the two environments (mostly) match the target phenotypes. From this, the authors conclude that DNA supercoiling can help shape both gene regulation and genome organization during evolution.

The manuscript is very clearly written and the analyses performed are well-described. I liked reading the manuscript, but I do have a number of more conceptual questions on the analyses performed:

- A major concern I have is that the results presented may be evident given the model used. Given a model with a set of hard-coded rules, e.g. expression of divergently oriented genes induces negative supercoiling and negative supercoiling induces expression (positive feedback loop), only one influence on gene expression (supercoiling) and only one evolutionary operator (gene inversion), it is rather evident to me that evolution will use the inversion operator to position two AB genes that need to be always expressed (i.e. in environments A and B) in a divergent orientation. Even if the supercoiling-transcription model and the rule set used are true, the recovery of divergent AB gene pairs may be less evident if other regulatory influences (e.g. TFs) and other mutational operators (e.g. mutations in promoter regions) were taken into account. At the very least, potential modeling biases should be addressed in the discussion.

- Why are the genomes adapted to two different environments simultaneously instead of one ? It is mentioned at some point that these are two environments that the bacteria may encounter during their life cycle, but it would be helpful to have concrete examples, in particular for obligate endosymbionts with low numbers of transcription factors such as B. aphidicola, for which supercoiling-based transcriptional regulation may be most relevant.

- The authors state that different models of transcription-supercoiling coupling produce contrasting results. Yet they use only one model, the one of El Houdaigui et al., 2019, in their simulations. Could the results qualitatively change when another model is used, or, the other way around, could matching the evolved genome organizational characteristics of different models with real genome characteristics support some model and rule out others ? E.g., are motifs observed in the simulation results, such as divergent pairs of always-on genes and convergent gene pairs with toggle switch-like behavior, also observed in reality ? More links to reality would definitely help to increase confidence in the biological relevance of the simulation results.

- The authors consider a gene on/off if its expression is above/below the

expression threshold e1/2, which is the average between minimum and maximum expression. However, the actual evolutionary optima are minimum expression (e^-m) or maximum expression (1). Judging from Figure 4, the evolved populations are rather far away from these optima after 1 million generations of evolution. How does this relate to the power of supercoiling to influence gene expression, what would e.g. be needed to really shut off a gene (and is it even possible ?).

Minor comments:

- *σ*_0_ on line 704 is defined as the level of supercoiling at which the opening free energy is at half its maximum level. Is this the same thing as the parameter *σ*_*opt*_ defined in Table 1 as the promoter opening threshold ?

- Parameter ε in equation 3 should be defined, it is linked in Table 1 to ‘crossover width’ but it’s unclear what this means.

- The authors state on line 545 in the discussion that ‘the distribution of gene triplets in evolved genomes is not uniform, but is enriched in configurations that are theoretically predicted to be beneficial in the model.’. I don’t see a lot of solid theoretical predictions in the text. E.g. the authors state on line 359 that ‘Triplet (2) shows how an AB gene in a strongly-expressed divergent ←AB − AB→ pair can inhibit a B gene, and the B gene of the triplet can also be seen as reinforcing the expression of the divergent ←AB − AB→ pair in environment B.’, but reinforcing this divergent pair would lead to further inhibition of the B gene in the environment B where it is supposed to be expressed ? Couldn’t it be that the enrichment of some triplets is just a consequence of the enrichment of some of the underlying pairs such as ←AB − AB→ ?

- Figure 3: why are there already more activated AB and A genes than B genes in generation 1 of the adaptation to environment A, and more AB and B genes than A genes in generation 1 of the adaptation to environment B ? Is generation 1 the start point or is this the population after 1 generation of evolution (as I suspect) ? In the latter case, the horizontal trajectories before generation 1 do not help interpretation.

**Have the authors made all data and (if applicable) computational code underlying the findings in their manuscript fully available?**

Reviewer #1: Yes

Reviewer #2: Yes

Reviewer #3: Yes

PLOS authors have the option to publish the peer review history of their article (what does this mean?). If published, this will include your full peer review and any attached files.

Reviewer #1: No

Reviewer #2: No

Reviewer #3: No

**Figure resubmission:**
---

## [Decision Letter · Decision Letter 1]

3 Sep 2025

Dear Pr. Beslon,

We are pleased to inform you that your manuscript 'Emergence of Supercoiling-Mediated Regulatory Networks through the Evolution of Bacterial Chromosome Organization' has been provisionally accepted for publication in PLOS Computational Biology.

Best regards,

Tommy Tsan-Yuk Lam, Ph.D.

Academic Editor

PLOS Computational Biology

Stacey Finley, Ph.D.

Section Editor

PLOS Computational Biology

**Comments to the Authors:**

Reviewer #1: As noted in my previous report, the simulation results presented in this study largely reflect outcomes that would be expected from the assumptions built into the model, making it difficult to recognize strong novelty in this aspect. To demonstrate the value of the study, it is important to discuss how the results relate to experimental data. In the revised manuscript, the authors have summarized potentially relevant experimental data in Table 1 and have discussed their relationship to the theoretical findings. This effort is commendable. However, directly linking the theoretical results to experimental data remains a task for future work, and unfortunately, the study has not yet reached the stage of experimental validation.

Reviewer #2: The authors have well addressed my comments and much improved the manuscript. I recommend acceptance.

Reviewer #3: The authors satisfactorily addressed my comments. The only thing I think should be mentioned more explicitly in the discussion/conclusion is that the relative functional and evolutionary importance of supercoiling versus other mechanisms of gene expression regulation (TFs, epigenetic modifications) is still unclear and remains to be investigated in more detail.

**Have the authors made all data and (if applicable) computational code underlying the findings in their manuscript fully available?**

Reviewer #1: None

Reviewer #2: Yes

Reviewer #3: Yes

PLOS authors have the option to publish the peer review history of their article (what does this mean?). If published, this will include your full peer review and any attached files.

Reviewer #1: No

Reviewer #2: No

Reviewer #3: No

---

## [Editor Report · Acceptance letter]

PCOMPBIOL-D-24-02197R1

Emergence of Supercoiling-Mediated Regulatory Networks through the Evolution of Bacterial Chromosome Organization

Dear Dr Beslon,

I am pleased to inform you that your manuscript has been formally accepted for publication in PLOS Computational Biology. Your manuscript is now with our production department and you will be notified of the publication date in due course.

With kind regards,

Anita Estes
